Cranial osteology, taxonomic reassessment, and phylogenetic relationships of the Early Cretaceous (Aptian-Albian) turtle Trinitichelys hiatti (Paracryptodira)

Rollot Yann 1 yann.rollot@gmail.com
http://orcid.org/0000-0002-2393-5621 Evers Serjoscha W. 1
Pierce Stephanie E. 2
Joyce Walter G. 1
1 Department of Geosciences, University of Fribourg , Fribourg , Switzerland
2 Department of Organismic and Evolutionary Biology, Museum of Comparative Zoology , Cambridge, Massachusetts , United States of America
Knoll Fabien
Electronic publication date: 2022 Nov 2
Publication date: 2022
Volume: 10
Electronic Location ID: e14138
Received 2022 Mar 21; Accepted 2022 Sep 6
Copyright: © 2022 Rollot et al.
Copyright year: 2022
Copyright holder: Rollot et al.
License: This is an open access article distributed under the terms of the Creative Commons Attribution License, which permits unrestricted use, distribution, reproduction and adaptation in any medium and for any purpose provided that it is properly attributed. For attribution, the original author(s), title, publication source (PeerJ) and either DOI or URL of the article must be cited.
License URL: https://creativecommons.org/licenses/by/4.0/

Keywords: Testudinata, Paracryptodira, Baenidae, Anatomy, Systematics, Phylogeny, Turtles

Funding: Swiss National Science Foundation SNF 200021_178780/1 Yann Rollot, Serjoscha Evers, and Walter Joyce received funding from the Swiss National Science Foundation via project (SNF 200021_178780/1). There was no additional external funding received for this study. The funders had no role in study design, data collection and analysis, decision to publish, or preparation of the manuscript.

==============================
We describe the skull of the Early Cretaceous (Aptian-Albian) baenid turtle Trinitichelys hiatti using micro-computed tomography to provide new insights into the cranial anatomy of basal baenids and into the evolution of paracryptodires. We show that the validity of Trinitichelys hiatti vs Arundelemys dardeni still holds true, that the most basal known baenids for which skull material is known share an intriguing combination of features that are typical of either Pleurosternidae or Baenidae, and that the carotid system of Trinitichelys hiatti is intermediate to that of pleurosternids and more advanced baenids. Our expanded phylogenetic analysis confirms the traditional placement of Arundelemys dardeni, Lakotemys australodakotensis, and Trinitichelys hiatti as basal baenids, retrieves Helochelydridae along the stem of Baenoidea, but recovers Dinochelys whitei, Glyptops ornatus, Dorsetochelys typocardium, and Uluops uluops as basal branching Paracryptodira.

Introduction

Baenidae Cope, 1873 is a diverse clade of aquatic paracryptodiran turtles that lived in North America from the Early Cretaceous to the Eocene (Hay, 1908; Gaffney, 1972; Joyce & Lyson, 2015). The clade includes all turtles more closely related to Baena arenosa Leidy, 1870 than Pleurosternon bullockii (Owen, 1842) or any extant turtle (Lyson & Joyce, 2011). Along with Pleurosternidae, Baenidae is included in the clades Baenoidea Williams, 1950 and Paracryptodira Gaffney, 1975, which are defined as the least inclusive and most inclusive clades containing Baena arenosa and Pleurosternon bullockii (Lyson & Joyce, 2011), respectively. Additional groups of European and North American turtles have been found to have paracryptodiran affinities, such as Compsemydidae Pérez-García, Royo-Torres & Cobos, 2015, which have either been retrieved as basal branching paracryptodires (Pérez-García, Royo-Torres & Cobos, 2015) or baenids (Rollot, Evers & Joyce, 2021a), and Helochelydridae Nopcsa, 1928, which were found to be pleurosternids (Rollot, Evers & Joyce, 2021a).

Baenidae is particularly diverse from the Santonian to the Eocene (Joyce & Lyson, 2015), but little is known about its most basal representatives in the Early Cretaceous. Only four taxa are currently recognized in the Early Cretaceous fossil record as valid baenids: Arundelemys dardeni Lipka et al., 2006 from the Aptian–Albian of Maryland, Lakotemys australodakotensis Joyce, Rollot & Cifelli, 2020 from the Berriasian–Valanginian, Protobaena wyomingensis (Gilmore, 1920) from the Albian of Wyoming, and Trinitichelys hiatti Gaffney, 1972 from the Aptian–Albian of Texas. New insights have recently been provided into the cranial anatomy of Arundelemys dardeni (Evers, Rollot & Joyce, 2021) and Lakotemys australodakotensis (Rollot et al., 2022), but many uncertainties remain about the anatomy of Protobaena wyomingensis and Trinitichelys hiatti. Trinitichelys hiatti is exclusively known from the holotype, MCZ VPRA-4070, a nearly complete skeleton that only lacks the mandible, the posterior part of the shell, the caudal vertebrae, and the distal parts of the limbs. A brief diagnosis of the species along with a few illustrations and reconstructions of the skull and shell were originally provided by Gaffney (1972), but a detailed description of the holotype is still lacking. Trinitichelys hiatti has, nevertheless, regularly been included in phylogenetic analyses of Paracryptodira and always is retrieved as one of the most basal baenids (Lyson & Joyce, 2009a, 2009b, 2010, 2011; Pérez-García, Royo-Torres & Cobos, 2015; Pérez-García et al., 2015; Lyson et al., 2016, 2021; Lyson, Sayler & Joyce, 2019; Joyce & Rollot, 2020).

As part of a project that aims to provide detailed insights into the cranial anatomy of basal baenids and better understand the evolution of paracryptodires, we provide a complete description of the skull of the holotype of Trinitichelys hiatti based on µCT scans. We compare our interpretation of the cranial osteology of Trinitichelys hiatti to the original work of Gaffney (1972) and discuss similarities with Arundelemys dardeni and Lakotemys australodakotensis, as well as the validity of those taxa. We also provide an expanded phylogenetic analysis of paracryptodires that takes into account the latest discoveries and insights into paracryptodiran anatomy and relationships.

Materials and Methods

The skull of MCZ VPRA-4070 was scanned using a Bruker SkyScan 1173 at the Museum of Comparative Zoology (MCZ) Digital Imaging Facility, Harvard University. The scan was performed with a voltage of 130 kV, a current of 60 µA, and a 0.25 mm brass filter. A total of 2,206 coronal slices were obtained with a voxel size of 29.2 µm. The specimen was segmented using the software Amira (version 2021.2; https://www.fei.com/software). Bone-by-bone reconstructions were obtained through interpolated slice-by-slice segmentations and the 3D models were exported as .ply files. The software Blender 2.79b (https://www.blender.org) was used to create the images for the figures. The µCT slice data and 3D models are archived on MorphoSource under the MCZ Vertebrate Paleontology Organization (https://www.morphosource.org/projects/000417478?).

The phylogenetic relationships of Trinitichelys hiatti were investigated by using the taxon-character matrix of Rollot, Evers & Joyce (2021a). The matrix was expanded to include the meiolaniformes Peligrochelys walshae from the Paleocene of Argentina (Sterli & de la Fuente, 2013, 2019) and Chubutemys copelloi from the Aptian–Albian of Argentina (Gaffney et al., 2007; Sterli, de la Fuente & Umazano, 2015); the sichuanchelyids Mongolochelys efremovi from the Maastrichtian of Mongolia (Sukhanov, 2000; Suzuki & Chinzorig, 2010) and Sichuanchelys palatodentata from the Oxfordian of China (Joyce et al., 2016); the indeterminate early stem turtle Eileanchelys waldmani from the Bathonian of Scotland, UK (Anquetin et al., 2009; Anquetin, 2010); the recently described pleurosternid Pleurosternon moncayensis from the Tithonian-Berriasian of Spain (Pérez-García et al., 2021); and the baenids Lakotemys australodakotensis from the Berriasian-Valanginian of South Dakota, USA (Rollot et al., 2022), Neurankylus torrejonensis from the Paleocene of New Mexico, USA (Lyson et al., 2016), Goleremys mckennai from the Paleocene of California, USA (Hutchison, 2004), Saxochelys gilberti from the Maastrichtian of North Dakota, USA (Lyson, Sayler & Joyce, 2019), and Palatobaena knellerorum from the Paleocene of Colorado USA (Lyson et al., 2021). Although not a global matrix with an exhaustive taxon set, our taxon additions ensure that critical groups of turtles to test the content of Paracryptodira are included. In particular, the addition of early diverging turtle species and clades such as Meiolaniformes provide tests for the controversial phylogenetic positions of Kallokibotion bajazidi, compsemydids and helochelydrids, which have variously been found as paracryptodires or not (e.g., Joyce et al., 2016; Pérez-García & Codrea, 2018; Rollot, Evers & Joyce, 2021a).

In addition to taxonomic expansion, three characters were modified (characters 9, 11, and 26; see Supplemental Information) and two were deleted (characters 5 and 25). A further 22 characters were run ordered, because they form morphoclines (characters 5, 9, 13, 15, 17, 25, 26, 29, 32, 37, 38, 39, 44, 46, 58, 61, 78, 86, 93, 95, 96, and 99). The character scoring for Arundelemys dardeni and Trinitichelys hiatti were updated following the latest insights into the cranial anatomy of these taxa (Evers, Rollot & Joyce, 2021; this study; see Supplemental Information). The scoring for Pleurosternon bullockii were updated following recent investigation of morphological variability in the shell of this taxon (Guerrero & Pérez-García, 2020, 2021; see Supplemental Information) and additional changes were implemented for Compsemys victa, Peckemys brinkman, Helochelydra nopcsai, Naomichelys speciosa, and Kallokibotion bajazidi (see Supplemental Information).

The final matrix included 48 species and 105 characters and was subjected to a traditional parsimony analysis using TNT (version 1.5; Goloboff, Farris & Nixon, 2008). The first analysis was performed under equal weighting and the second analysis was carried out with the implementation of an implied weighting factor of K = 12, following recommendations from Goloboff, Torres & Arias (2018). One thousand random addition sequences were followed by a round of tree bisection reconnection. Trees suboptimal by 10 steps and with a relative fit difference of 0.1 were retained as part of the first search. A tree collapsing rule was implemented with a minimum length of 0. Proganochelys quenstedti was chosen as the outgroup.

SYSTEMATIC PALAEONTOLOGY

TESTUDINATA Klein, 1760 [Joyce et al., 2020]

PARACRYPTODIRA Gaffney, 1975 [Joyce et al., 2021]

BAENOIDEA Williams, 1950 [Joyce et al., 2021]

BAENIDAE Cope, 1873 [Joyce et al., 2021]

TRINITICHELYS Gaffney, 1972

Trinitichelys hiatti Gaffney, 1972

Holotype: MCZ VPRA-4070, a partial skeleton, only lacking the mandible, the caudal vertebrae, the distal parts of the limbs, and the posterior part of the shell (Gaffney, 1972, figs. 2–5, 47; Gaffney, 1982, fig. 3; Figs. 1–7).

Figure 1 Skull of Trinitichelys hiatti (MCZ VPRA-4070, holotype), Early Cretaceous (Aptian-Albian) of Texas, U.S.A.

Three-dimensional renderings of the skull and illustrations in (A) dorsal view, (B) ventral view, (C) anterior view, (D) posterior view, and (E) right lateral view. Abbreviations: bo, basioccipital; ex, exoccipital; fon, foramen orbito-nasale; fpp, foramen palatinum posterius; fpr, foramen praepalatinum; fr, frontal; fst, foramen stapedio-temporale; ica, incisura columella auris; j-qj, jugal-quadratojugal; mx, maxilla; na, nasal; op, opisthotic; pa, parietal; pal, palatine; pbs, parabasisphenoid; pf, prefrontal; pmx, premaxilla; po, postorbital; pro, prootic; pt, pterygoid; qu, quadrate; so, supraoccipital; sq, squamosal; tb, tuberculum basioccipitale; vo, vomer.

Figure 2 Cranial scutes of Trinitichelys hiatti (MCZ VPRA-4070, holotype), Early Cretaceous (Aptian-Albian) of Texas, U.S.A.

Three-dimensional renderings of the skull and illustrations in (A) dorsal view and (B) right lateral view. Sutural lines are thin black lines, whereas thick lines are scute sulci, which are labeled as capital labels.

Figure 3 Three-dimensional renderings of the segmented right side of the skull of Trinitichelys hiatti (MCZ VPRA-4070, holotype).

(A) Right lateral view with the jugal-quadratojugal removed, (B) right medial view. Abbreviations: ?b, unidentified piece of bone; bo, basioccipital; ex, exoccipital; fio, foramen interorbitale; fon, foramen orbito-nasale; fpp, foramen palatinum posterius; fr, frontal; ica, incisura columella auris; mx, maxilla; na, nasal; co, condylus occipitalis; pa, parietal; pal, palatine; pbs, parabasisphenoid; pf, prefrontal; pmx, premaxilla; po, postorbital; ppe, processus pterygoideus externus; pro, prootic; pt, pterygoid; qu, quadrate; so, supraoccipital; sq, squamosal; ‘tf’, putative location of the trigeminal foramen; vo, vomer.

Figure 4 Three-dimensional renderings of the bones from the braincase and palatal regions of the skull of Trinitichelys hiatti (MCZ VPRA-4070, holotype).

(A) Anterolateral view of the right parietal, (B) medial view of the right premaxilla, (C) ventral view of the right premaxilla, and (D) ventral view of the right palatine. Abbreviations: dppa, descending process of the parietal; fon, foramen orbito-nasale; fpp, foramen palatinum posterius; fpr, foramen praepalatinum; lr, labial ridge; pa, parietal; pal, palatine; pmx, premaxilla; r-spp, ridge posteriorly framing the sulcus palatino-pterygoideus; s-fr, suture with the frontal; s-max, suture with the maxilla; s-pmx, suture with the premaxilla; s-po, suture with the postorbital; s-vo, suture with the vomer.

Figure 5 Three-dimensional renderings of the right pterygoid of the skull of Trinitichelys hiatti (MCZ VPRA-4070, holotype).

(A) Ventral view, (B) dorsal view, (C) dorsolateral view, and (D) posterolateroventral view. Abbreviations: app, anterior process of the pterygoid; bo-f, articulation facet with the basioccipital; ccv, canalis cavernosus; cp, carotid pit; crpt, crista pterygoidei; fdnv, foramen distalis nervi vidiani; fpcnv, foramen posterius canalis nervi vidiani; fpnv, foramen proximalis nervi vidiani; j-f, articulation facet with the jugal; pal-f, articulation facet with the palatine; pbs-f, articulation facet with the parabasisphenoid; pfo, pterygoid fossa; ppe, processus pterygoideus externus; ppp, posterior process of the pterygoid; pro-f, articulation facet with the prootic; qu-f, articulation facet with the quadrate; scv, sulcus cavernosus; vf, vertical flange of the processus pterygoideus externus.

Figure 6 Three-dimensional renderings of the parabasisphenoid and right pterygoid of the skull of Trinitichelys hiatti (MCZ VPRA-4070, holotype).

(A) Dorsal view, (B) dorsal view of the bones rendered transparent showing the internal carotid artery and facial nerve systems, (C) ventral view, and (D) ventral view of the bones rendered transparent showing the internal carotid artery and facial nerve systems. Abbreviations: ccb, canalis caroticus basisphenoidalis; ccv, canalis cavernosus; cg, carotid groove; cna, canalis nervus abducentis; cnv, canalis nervus vidianus; cp, carotid pit; cprnv, canalis pro ramo nervi vidiani; ds, dorsum sellae; faccb, foramen anterius canalis carotici basisphenoidalis; facna, foramen anterius canalis nervi abducentis; facnv, foramen anterius canalis nervi vidiani; fpccb, foramen posterius canalis carotici basisphenoidalis; fpcna, foramen posterius canalis nervi abducentis; fpcnv, foramen posterius canalis nervi vidiani; fpnv, foramen proximalis nervi vidiani; ica, internal carotid artery; le-pbs, lateral extension of the parabasisphenoid; me-pt, medial extension of the pterygoid; pbs, parabasisphenoid; pt, pterygoid; rbp, retractor bulbi pit; scv, sulcus cavernosus; st, sella turcica.

Figure 7 Three-dimensional renderings of the right ceratobranchial I of Trinitichelys hiatti (MCZ VPRA-4070, holotype).

(A) Dorsal view, (B) ventral view, (C) medial view, and (D) lateral view.

Type locality and horizon: 1 mile from Hardee, on road to Forestburg, Montague County, Texas, USA. (Gaffney, 1972); Antlers Formation (Trinity Sands of Gaffney, 1972), Aptian-Albian, Early Cretaceous (Joyce, Sterli & Chapman, 2014).

Revised diagnosis: Trinitichelys hiatti can be diagnosed as a baenid by a reduced, only anteriorly developed lingual ridge, a ventral extension of the jugal, a deep upper temporal emargination, a well-developed posterior process of the pterygoid with an extensive contact with the basioccipital, absence of basipterygoid processes, absence of a second pair of anterior tubercula basioccipitale, presence of axillary buttresses with an extensive contact with the costals, presence of extensive inguinal buttresses, and the absence of epiplastral processes. Trinitichelys hiatti is closer to Lakotemys australodakotensis than any other baenid and shares with the latter an elongate skull, a small nasal, and a maximum combined width of the parietals greater than their length. Trinitichelys hiatti can be differentiated from all other baenids by the following combination of features: presence of a carotid pit, an articular surface of the condylus occipitalis formed by the basioccipital only, a midline contact of the pterygoids for 40–70% of their length, a pleurosternid-like skull sculpturing, nasals that extend as far anteriorly as the premaxillae, vertebrals significantly wider than long, vertebral III narrower anteriorly than posteriorly, gulars smaller than extragulars.

Referred material: none.

Description

Skull. The skull of MCZ VPRA-4070 is dorsoventrally crushed and diagonally distorted (Fig. 1). Most palatal and braincase elements are damaged but the cheek area and skull roof bones are overall well preserved. Damage to both otic capsules prevents any clear observation of their internal anatomy. The cavum labyrinthicum could not be reconstructed. The right side of MCZ VPRA-4070 is less affected by crushing than the left side, which allows for detailed descriptions of almost every bone on this side. However, we are not able to rigorously differentiate the jugal from the quadratojugal. These two bones are, therefore, left unresolved in the figures and models (Fig. 1E).

The skull of MCZ VPRA-4070 resembles Pleurosternon bullockii (Evers, Rollot & Joyce, 2020) by being longer than wide (Fig. 1A) but still appears to be less elongate than Glyptops ornatus (Glyptops plicatulus of Gaffney, 1979) and differs from the wedge-shaped skull of baenodds (Joyce & Lyson, 2015). The approximate length between the anterior tip of the premaxillae and the condylus occipitalis is 54 mm, and the approximate width between the squamosals dorsal to the cavum tympani is 44 mm. The skull surface of Trinitichelys hiatti is sculptured by low and irregular pits that are particularly well preserved on the skull roof and the lateral surface of the maxillae. The surface texture is similar to Arundelemys dardeni (Lipka et al., 2006; Evers, Rollot & Joyce, 2021) but differs from the irregular tubercles of Glyptops ornatus (Gaffney, 1979), Pleurosternon bullockii (Evers, Rollot & Joyce, 2020), Pleurosternon moncayensis (Pérez-García et al., 2021), and Uluops uluops (Rollot, Evers & Joyce, 2021a), or the fine crenulations in Dorsetochelys typocardium (DORCM G 23). The surface texture along the posterior portion of the parietals of Trinitichelys hiatti shows a slightly more striated pattern, albeit less pronounced than the ridge-like arrangement exhibited in the same area in Pleurosternon bullockii (Evers, Rollot & Joyce, 2020).

Only a few scute sulci are preserved on the right side of the skull roof (see Fig. 2), but those do not allow reconstruction of the cranial scute pattern. Following the scute nomenclatural system of Sterli & de la Fuente (2013), we tentatively identify some cranial scutes by comparing them to the scute pattern of Arundelemys dardeni (Evers, Rollot & Joyce, 2021) and Eubaena cephalica (Rollot, Lyson & Joyce, 2018). The dorsal process of the prefrontal is mediolaterally crossed by a sulcus that separates scute Z anteriorly from scute Y posteriorly (Fig. 2). The second sulcus can be traced from the posterior aspect of the orbit margin, extending posteriorly on the postorbital, before crossing the parietal transversally just posterior to the frontal-parietal suture (Fig. 2). The sulcus separates scute F2 from scute D posteriorly and F3 laterally, respectively. The third sulcus is located along the posterior part of the parietal, which it crosses sagittally to reach the posterior skull roof margin, and we interpret this sulcus to separate scute A medially from scute H laterally (Fig. 2A).

Despite shearing to the skull, the orbits can be interpreted as being vertically oriented (Figs. 1A, 1C), which resembles the condition observed in Compsemys victa (Lyson & Joyce, 2011), Arundelemys dardeni (Lipka et al., 2006; Evers, Rollot & Joyce, 2021), Lakotemys australodakotensis (Rollot et al., 2022), and Saxochelys gilberti (Lyson, Sayler & Joyce, 2019) but contrasts with Eubaena cephalica (Rollot, Lyson & Joyce, 2018), Palatobaena cohen (Lyson & Joyce, 2009a), Palatobaena knellerorum (Lyson et al., 2021), and Cedrobaena putorius (Lyson & Joyce, 2009b). The cheek emargination is moderately deep, formed by the maxilla, jugal, and quadratojugal, and just reaches the level of the lower margin of the orbit (Fig. 1E). The cheek emargination of MCZ VPRA-4070 is slightly less deep than that of Pleurosternon bullockii (Evers, Rollot & Joyce, 2020), Uluops uluops (Rollot, Evers & Joyce, 2021a), and Arvinachelys goldeni (Lively, 2015), but overall resembles that of Lakotemys australodakotensis (Rollot et al., 2022) and the baenodds Plesiobaena antiqua (Brinkman, 2003), Chisternon undatum (Gaffney, 1972), Eubaena cephalica (Rollot, Lyson & Joyce, 2018). A well-developed processus pterygoideus externus with a large vertical flange is present as in Pleurosternon bullockii (Evers, Rollot & Joyce, 2020), Uluops uluops (Rollot, Evers & Joyce, 2021a), Arundelemys dardeni (Lipka et al., 2006; Evers, Rollot & Joyce, 2021), and Lakotemys australodakotensis (Rollot et al., 2022) which differs from the reduced processus pterygoideus externus of most baenodds.

The upper temporal emargination is relatively deep with the foramen stapedio-temporale being exposed in dorsal view, but the anterior portion of the otic capsule remains covered by the skull roof (Fig. 1A). In lateral view, the upper temporal emargination extends to the level of the anterior margin of the cavum tympani. A similarly deep upper temporal emargination is found in Lakotemys australodakotensis (Rollot et al., 2022) and baenodds (Gaffney, 1972; Joyce & Lyson, 2015), with the exception of Baena arenosa (Gaffney, 1972), and contrasts with the shallower upper temporal emargination of non-baenodds, such as Dorsetochelys typocardium (Evans & Kemp, 1976), Neurankylus torrejonensis (Lyson et al., 2016), Pleurosternon bullockii (Evers, Rollot & Joyce, 2020), or Uluops uluops (Rollot, Evers & Joyce, 2021a).

Nasal. The nasal is moderately large and, despite damage that does not allow its general shape to be assessed, appears to be wider than long (Fig. 1A). The nasal forms the dorsal margin of the apertura narium externa and roofs the nasal cavity. Gaffney (1972) illustrated the nasal of Trinitichelys hiatti as being relatively large and having a pointed contribution to the orbit, but we find the nasal to be much smaller and to lack any contribution to the orbit (Figs. 1A, 1E). Within the nasal cavity, the nasal contacts its counterpart medially and, along their medial aspect, they form a protruding ridge that divides the nasal cavity into left and right halves. In dorsal view, the nasal also contacts the anterior process of the frontal posteromedially and the dorsal plate of the prefrontal posterolaterally (Fig. 1A). The anterior process of the frontals only slightly protrudes between the nasals posteriorly and the nasals contact each other along the midline for most of their length (Fig. 1A), as in Compsemys victa (Lyson & Joyce, 2011), Dorsetochelys typocardium (Evans & Kemp, 1976), Glyptops ornatus (Gaffney, 1979), Uluops uluops (Rollot, Evers & Joyce, 2021a), and most baenodds (Joyce & Lyson, 2015), but unlike Pleurosternon bullockii (Evers, Rollot & Joyce, 2020), Arundelemys dardeni (Lipka et al., 2006; Evers, Rollot & Joyce, 2021), Lakotemys australodakotensis (Rollot et al., 2022) and Neurankylus torrejonensis (Lyson et al., 2016). The crushing and shearing of the anteriormost portion of the skull does not allow us to infer with confidence if a contact was present between the nasal and the maxilla, but our 3D reconstructions suggest that a nasal-maxilla contact is prevented on the right side by an unusual anterior prefrontal process that reaches the dorsolateral margin of the apertura narium externa (see prefrontal; Figs. 1A, 1C), as is the case in Lakotemys australodakotensis (Rollot et al., 2022), but we also note that this lack of a contact may be the result of damage to the anterior margin of the nasal. The left side is not informative in this regard as its morphology is obscured by damage. Almost all paracryptodires with distinct nasals exhibit a nasal-maxilla contact along the anteriormost aspect of the skull (Gaffney, 1972; Evans & Kemp, 1975, 1976; Lyson & Joyce, 2011; Joyce & Lyson, 2015; Lively, 2015; Rollot, Evers & Joyce, 2021a), with the exception of Naomichelys speciosa and Helochelydra nopcsai, where the prefrontal prevents the two bones from contacting each other (Joyce et al., 2011; Joyce, Sterli & Chapman, 2014). This contact may be absent in numerous baenodds as well (e.g., Chisternon undatum, Gaffney, 1972), but difficult to assess, given that the nasal is often fused to the frontal (Gaffney, 1972).

Prefrontal. The prefrontal forms the anterodorsal margin of the orbit and perhaps contributes to the posterolateral margin of the external nares (Figs. 1A, 1E, and 3A). The dorsal plate of the prefrontal is clearly exposed on the skull roof but is prevented from contacting its counterpart by the anterior process of the frontal that reaches the nasal (Fig. 1A). This strongly contrasts with the original description of Gaffney (1972), which asserts that the prefrontal does not contribute to the dorsal skull roof at all. An exposure of the prefrontal on the skull roof dorsal to the orbit is generally present among paracryptodires, including Arundelemys dardeni (Lipka et al., 2006; Evers, Rollot & Joyce, 2021), Lakotemys australodakotensis (Rollot et al., 2022), Compsemys victa (Lyson & Joyce, 2011), Dorsetochelys typocardium (Evans & Kemp, 1976), Arvinachelys goldeni (Lively, 2015), Neurankylus eximius (Brinkman & Nicholls, 1993), Neurankylus torrejonensis (Lyson et al., 2016), Pleurosternon bullockii (Evers, Rollot & Joyce, 2020), or Uluops uluops (Rollot, Evers & Joyce, 2021a), but is greatly reduced to absent in baenodds (Brinkman & Nicholls, 1991; Joyce & Lyson, 2015). The dorsal plate of the prefrontal bears an anterior process that anteriorly inserts between the nasal and the maxilla. Although our 3D models show that this process prevents the nasal from contacting the maxilla, it is unclear to us if this is genuine (see nasal). The dorsal plate of the prefrontal contacts the nasal anteromedially, the maxilla anterolaterally, and the frontal posteromedially and posteriorly (Figs. 1A, 1E, and 3A). The descending process of the prefrontal forms the anteromedial wall of the fossa orbitalis (Figs. 1E and 3A). Despite damage that affects the ventral portions of the prefrontal and vomer, the arrangement of these bones in that area allows us to suggest that the descending process likely contributes to the margin of the enlarged foramen orbito-nasale. Within the orbit, the prefrontal-frontal contact forms a W-shaped suture in ventral view, which reminds of the condition observed in Uluops uluops (Rollot, Evers & Joyce, 2021a), albeit weaker interdigitated as in the latter. The descending process of the prefrontal contacts the ascending process of the maxilla anteriorly and laterally and the vomer ventromedially (Figs. 1E and 3A). A contact between the descending process of the prefrontal and the palatine within the fossa orbitalis is not preserved as is, but the interpolation of the intact margins of the foramen orbito-nasale suggests that this contact was present, as in the vast majority of turtles. The descending process of the prefrontal of Trinitichelys hiatti seems to slightly extend medially within the nasal cavity, but a sheet-like ridge, as identified in Arundelemys dardeni (Evers, Rollot & Joyce, 2021) and Lakotemys australodakotensis (Rollot et al., 2022), is not apparent.

Frontal. The frontal is broad posteriorly and abruptly narrows anteriorly by way of a process that starts anterior to its orbital contribution and extends medially to the dorsal process of the prefrontal to reach the nasal (Fig. 1A). The frontal is about twice as long anteroposteriorly as it is wide mediolaterally at its posterior contact with the parietal. The frontal forms a short process at about mid-length that extends laterally to contribute to the dorsal margin of the orbit. This process inserts between the dorsal process of the prefrontal anteriorly and the postorbital posteriorly, thus preventing both bones from contacting each other (Figs. 1A, 1C, 1E, and 3A). A frontal contribution to the orbit margin is present in a broad selection of paracryptodires, including Arundelemys dardeni (Lipka et al., 2006, Evers, Rollot & Joyce, 2021), Lakotemys australodakotensis (Rollot et al., 2022), Dorsetochelys typocardium (Evans & Kemp, 1976), Helochelydra nopcsai (Joyce et al., 2011), Naomichelys speciosa (Joyce, Sterli & Chapman, 2014), Neurankylus torrejonensis (Lyson et al., 2016), Pleurosternon bullockii (Evers, Rollot & Joyce, 2020), Uluops uluops (Rollot, Evers & Joyce, 2021a), and most baenodds (Gaffney, 1982), but is notably absent in Arvinachelys goldeni (Lively, 2015), Compsemys victa (Lyson & Joyce, 2011), Hayemys latifrons (Gaffney, 1972), and Gamerabaena sonsalla (Lyson & Joyce, 2010). The anterior process of the frontal represents about half of its length and prevents the prefrontals from contacting one another along the midline of the skull (Fig. 1A). The anterior process contacts the nasal anteriorly and the prefrontal anterolaterally and laterally (Fig. 1A). The frontal otherwise contacts its counterpart medially along its full length, the parietal posteriorly, and the postorbital posterolaterally (Figs. 1A, 1C, 1E and 3). Each frontal ventrally bears a poorly developed crista cranii, which collectively form the shallow sulcus olfactorius. The crista cranii level off posteriorly on the underside of the frontal and are not directly continuous with the margin of the descending process of the parietal, as is also the case in Arundelemys dardeni (Evers, Rollot & Joyce, 2021), Lakotemys australodakotensis (Rollot et al., 2022), Pleurosternon bullockii (Evers, Rollot & Joyce, 2020), and Uluops uluops (Rollot, Evers & Joyce, 2021a). The overall shape of the frontal differs significantly from the original description of Gaffney (1972) who illustrated it to be a rectangular bone that has a straight anterior suture with the nasal and that contributes broadly to the margin of the orbit.

Parietal. The parietal is about twice as long as wide and forms the medial and posterior portions of the skull roof (Fig. 1A). Its shape and contacts broadly conform to the skull reconstructions provided by Gaffney (1972). The dorsal plate of the parietal roofs the braincase and forms most of the upper temporal emargination (Figs. 1A and 3B), of which the extent is described above (see Skull). On the dorsal skull roof, the dorsal plate contacts the frontal anteriorly, the postorbital laterally, and the supraoccipital posteromedially (Figs. 1A, 1C, 1E, and 3). Although reduced, a clear contact of the parietal with the squamosal is visible along the posterolateral tip of the parietal on the right side of the skull (Fig. 1A), contra Gaffney (1972). A parietal-squamosal contact is known to be present in Lakotemys australodakotensis (Rollot et al., 2022), Dorsetochelys typocardium (Evans & Kemp, 1976), Pleurosternon bullockii (Evans & Kemp, 1975; Evers, Rollot & Joyce, 2020), Uluops uluops (Rollot, Evers & Joyce, 2021a), Helochelydra nopcsai (Joyce et al., 2011), Naomichelys speciosa (Joyce, Sterli & Chapman, 2014), and Neurankylus torrejonensis (Lyson et al., 2016). Among baenodds, it is generally absent, with exception of Baena arenosa and Chisternon undatum (Gaffney, 1972). The descending process of the parietal forms the posterior margin of the foramen interorbitale, the anterior part of the lateral wall of the cavum cranii, and the medial margin of the fossa temporalis (Fig. 3). The parietal slightly underlies the postorbital for a short distance at the base of the processus inferior parietalis. At the level of this underlying sheet of bone and slightly posterior to the ventral process of the postorbital, the parietal bears a low mediolateral ridge, which forms the posterior margin of the roof of the sulcus palatino-pterygoideus (Fig. 4A), as observed in Arundelemys dardeni (Evers, Rollot & Joyce, 2021), Lakotemys australodakotensis (Rollot et al., 2022), Pleurosternon bullockii (Evers, Rollot & Joyce, 2020), and Uluops uluops (Rollot, Evers & Joyce, 2021a). However, this feature is comparatively poorly developed with regard to the aforementioned taxa. As the ventral half of the processus inferior parietalis is strongly damaged on both sides, the bony contributions to the foramen nervi trigemini and the extent of the contact of the processus inferior parietalis with the pterygoid and purported epipterygoid cannot be assessed precisely (Figs. 3A and 4A). Within the upper temporal fossa, a broad contact with the prootic and a broad posterolateral contact with the supraoccipital can nevertheless be identified with confidence (Fig. 3A).

Postorbital. The postorbital is an elongate, plate-like bone that forms the posterior margin of the orbit and the dorsolateral aspect of the skull roof (Figs. 1A, 1C, 1E, and 3A). A mediolaterally thickened, ventral process is developed along the anterior portion of the postorbital that rests on the jugal and maxilla. Along with the medial process of the jugal, the ventral process of the postorbital forms the well-developed posterior wall of the fossa orbitalis that constricts the passage from the temporal fossa posteriorly to the orbital fossa anteriorly (Fig. 1C). The ventral process of the postorbital is thus very similar to that of Pleurosternon bullockii and Uluops uluops (Evers, Rollot & Joyce, 2020) and is reminiscent of the septum orbitotemporale of pleurodires (Gaffney, Tong & Meylan, 2006). Within the orbital fossa, the postorbital contacts the jugal ventromedially and the maxilla ventrolaterally. The medial process of the jugal prevents any contact of the postorbital with the palatine and the pterygoid. Along the skull roof, the width of the postorbital overall gradually decreases towards the posterior (Figs. 1A, 1E). The postorbital contacts the maxilla anteroventrally along the posteroventral margin of the orbit, which prevents the jugal from contributing to the latter (see jugal; Figs. 1E and 3A) and contrasts with the initial observations of Gaffney (1972), who highlighted a clear jugal contribution to the orbit. The postorbital contacts the frontal anteromedially, the parietal medially, the jugal anteroventrally along a sinusoid-shape suture, the quadratojugal posteroventrally, and the squamosal posteriorly (Figs. 1A, 1C, 1E, and 3A). A short contact between the parietal and squamosal posterior to the postorbital prevents the latter from contributing to the upper temporal emargination (Fig. 1A), as in Dorsetochelys typocardium (Evans & Kemp, 1976), Helochelydra nopcsai (Joyce et al., 2011), Naomichelys speciosa (Joyce, Sterli & Chapman, 2014), Pleurosternon bullockii (Evans & Kemp, 1975; Evers, Rollot & Joyce, 2020), and Uluops uluops (Rollot, Evers & Joyce, 2021a), but also the baenids Lakotemys australodakotensis (Rollot et al., 2022), Baena arenosa (Gaffney, 1972), Chisternon undatum (Gaffney, 1972), and Neurankylus torrejonensis (Lyson et al., 2016).

Jugal-quadratojugal. In the µCT data, we are not able to detect a clear suture between the jugal and the quadratojugal and thus segmented these two bones as a single mesh (Fig. 1E). The approximate extent of both bones can be assessed, with the exception of the posterior-most portion of the jugal and the anterior limit of the anterior process of the quadratojugal. Here, we figure and describe both bones together. The anterior third of this bony complex likely corresponds to the jugal, which forms the posteroventral part of the fossa orbitalis and the anterodorsal margin of the cheek emargination (Figs. 1C, 1E). The medial process of the jugal rests upon the posterior portion of the maxilla, and contacts the palatine medially, the pterygoid posteromedially, and the ventral process of the postorbital dorsally. Within the orbit, the medial process of the jugal contacts the maxilla along a V-shaped suture of which the anterior tip ends just posterior to the foramen supramaxillare (Fig. 1A). Along the lateral skull surface, the jugal contacts the maxilla anteroventrally, the postorbital anterodorsally and dorsally along a sinusoid-shape suture, and the quadratojugal posteriorly (Figs. 1A, 1C, and 1E). In his restoration, Gaffney (1972) suggested a broad contribution of the jugal to the orbit. Although the area of interest is crossed by various cracks, we are confident that the piece of bone originally identified as belonging to the jugal actually corresponds to the posterior part of the maxilla. The small contact now apparent between the postorbital and the maxilla in the posteroventral corner of the orbit margin thus prevents the jugal from contributing to the margin of the orbit (Fig. 1E), as in Arundelemys dardeni (Lipka et al., 2006; Evers, Rollot & Joyce, 2021), Lakotemys australodakotensis (Rollot et al., 2022), Glyptops ornatus (Gaffney, 1979), Pleurosternon bullockii (Evers, Rollot & Joyce, 2020), and a wide selection of baenodds (see Brinkman & Nicholls, 1991; Brinkman, 2003; Lyson & Joyce, 2010; Lively, 2015; Rollot, Lyson & Joyce, 2018; Lyson, Sayler & Joyce, 2019). The quadratojugal portion of the jugal-quadratojugal complex likely is a triradiate element that nearly forms the entire posterior margin of the cheek emargination (Fig. 1E). The posterior margin of the quadratojugal is concave, contacts the quadrate, but does not contribute to the anterior margin of the cavum tympani (Fig. 1E), confirming initial observations by Gaffney (1972). The dorsal process of the quadratojugal extends dorsally above the cavum tympani between the quadrate and postorbital to contact the squamosal and has a similar extent above the cavum tympani than most paracryptodires (Figs 1A, 1E), with the exception of some palatobaenines (Brinkman, 2003; Lyson & Joyce, 2009a, 2009b) and Compsemys victa (Lyson & Joyce, 2011), in which the process is shorter or absent. The posteroventral process extends ventrally along the anteroventral margin of the quadrate and almost reaches the condylus mandibularis (Fig. 1E). The anterior process contacts the postorbital dorsally and the jugal anteriorly (Figs. 1A, 1C, and 1E).

Squamosal. The squamosal is a cone-shaped element that forms the posterodorsolateral portion of the skull, the lateral margin of the upper temporal emargination, the posterodorsal margin of the cavum tympani, and most of the deep and voluminous antrum postoticum (Figs. 1A, 1D, 1E, and 3A). A broad concavity is developed along the ventrolateral margin of the posterior process of the squamosal. On the dorsal skull surface, the squamosal contacts the parietal anteromedially, the postorbital anteriorly, the quadratojugal anterolaterally, and the quadrate laterally (Figs. 1A, 1E, and 3A). Within the temporal fossa, the squamosal contacts the quadrate anteriorly and anteromedially and the paroccipital process of the opisthotic posteromedially (Figs. 1A, 1D). The dorsal margin of the squamosal forms the posterolateral margin of the upper temporal emargination. Its edge is medially curved to overhang the temporal fossa (Fig. 1A).

Premaxilla. The premaxilla forms the anteriormost part of the skull, ventrally contributes to the apertura narium externa and to the anterior aspect of the labial ridge, and floors the anteromedial portions of the fossa nasalis (Figs. 1B, 1C, 1E, 3, and 4B, 4C). The premaxilla forms a median ridge with its counterpart that slightly protrudes from below into the fossa nasalis and likely served as an insertion surface for the internarial septum. In ventral view, the premaxillae and vomer jointly form a “tongue groove,” a distinct depression at the front of the palate that is laterally framed by the triturating surfaces, and in which the anterior portion of the tongue is inferred to have been located (Fig. 1B). Among paracryptodires, such a tongue groove is otherwise developed in Arundelemys dardeni (Evers, Rollot & Joyce, 2021) and numerous baenodds (e.g., Lyson & Joyce, 2009a, 2009b; Rollot, Lyson & Joyce, 2018; Lyson, Sayler & Joyce, 2019). The premaxilla contacts its counterpart medially along the skull midline, the maxilla laterally, and the vomer posteriorly (Figs. 1B, 1C, 1E, and 3). The foramen praepalatinum is entirely formed by the premaxilla and is located at the posteriormost portion of the latter, very close to the suture with the vomer (Figs. 1B and 4C). A foramen praepalatinum exclusively formed by the premaxilla is also present in Arundelemys dardeni (Lipka et al., 2006), Dorsetochelys typocardium (DORCM G 23), Pleurosternon bullockii (Evans & Kemp, 1975; Evers, Rollot & Joyce, 2020), and a broad set of eubaenines (Gaffney, 1972), while a contribution from the vomer and/or maxilla is apparent in Arvinachelys goldeni (Lively, 2015) and palatobaenines (Gaffney, 1972; Archibald & Hutchison, 1979; Brinkman, 2003; Lyson & Joyce, 2009a, 2009b). The labial margin of the premaxilla is even and does not form a distinct hook (Fig. 1C) as seen in Compsemys victa (Lyson & Joyce, 2011).

Maxilla. The maxilla forms the lateral margins of the apertura narium externa, the anterolateral wall of the fossa nasalis, as well as the anteroventral and ventral aspects of the orbital margin and fossa orbitalis (Figs. 1A, 1C, 1E, and 3). The maxilla has an ascending process that extends dorsally to cover the lateral aspect of the descending process of the prefrontal (Figs. 1C, 1E, and 3A). The ascending process of the right maxilla is minorly exposed on the skull roof anterodorsal to the orbit, but its left counterpart is not apparent in dorsal view because of the damage and shearing that affects this area (Fig. 1A). Although such an extent for this process is highly unusual for a paracryptodire, we conclude that this interpretation is the one best supported by the µCT scans. The ascending process of the maxilla contacts the prefrontal dorsally and posteriorly, but contact with the nasal is barely prevented by the elongate anterior process of the prefrontal dorsal to the maxilla (Figs. 1A, 1C, 1E, and 3A). In lateral view, the maxilla forms the anteroventral margin of the cheek emargination and contacts the jugal posteriorly along an S-shaped suture and the postorbital posterodorsally (Figs. 1E and 3A). Within the fossa orbitalis, the maxilla forms the lateral margin of the relatively large foramen orbito-nasale and contacts the prefrontal anterodorsally, the vomer anteromedially, the palatine posterolaterally, and the jugal posteriorly (Figs. 1A, 1E, and 3A). The foramen supramaxillare is entirely formed by the maxilla and located just anterior to the V-shaped suture of the latter with the jugal (Fig. 1A). The canalis alveolaris superior can be traced for most of its length within the maxilla and the foramen alveolare superius is located medially at the base of the ascending process of the maxilla. The flat, but ventrodorsally rounded triturating surface is almost entirely formed by the maxilla, with extremely minor contributions from the premaxilla anteriorly and palatine laterally (Fig. 1B). The triturating surfaces anteriorly frame a tongue groove (see Premaxilla above). The triturating surface is relatively narrow anteriorly, but slightly expands posteriorly (Fig. 1B). Its dimensions are similar to that of Arundelemys dardeni (Lipka et al., 2006; Evers, Rollot & Joyce, 2021), Lakotemys australodakotensis (Rollot et al., 2022), Chisternon undatum (Gaffney, 1972), Arvinachelys goldeni (Lively, 2015), Pleurosternon bullockii (Evans & Kemp, 1975; Evers, Rollot & Joyce, 2020), and Uluops uluops (Rollot, Evers & Joyce, 2021a), but differs from broader surfaces of baenodds such as Eubaena cephalica (Rollot, Lyson & Joyce, 2018), Palatobaena spp. (Archibald & Hutchison, 1979; Lyson & Joyce, 2009a), Saxochelys gilberti (Lyson, Sayler & Joyce, 2019), and Stygiochelys estesi (Gaffney, 1972). The maxilla is bordered laterally by a high labial ridge, which forms a weak curve visible in lateral view along the anteroposterior length of the maxilla (Figs. 1E and 3A), as in Baena arenosa (Gaffney, 1972), Boremys pulchra (Brinkman & Nicholls, 1991), Arvinachelys goldeni (Lively, 2015), Compsemys victa (Lyson & Joyce, 2011), and Glyptops ornatus (Gaffney, 1979). The lingual ridge of Trinitichelys hiatti is low. The right maxilla has what seems to be a short accessory ridge near its contact with the premaxilla. However, a mirroring ridge is absent on the left side, so that it remains unclear if this ridge represents some abnormality or taphonomic feature, or a polymorphically developed accessory ridge.

Vomer. The vomer is an unpaired bone that forms the posteroventral part of the fossa nasalis, the anteromedial margin of the foramen orbito-nasale, and the medial margin of the apertura narium interna (Figs. 1B and 3). The anterior half of the vomer is significantly broadened between the maxillae while the posterior half is narrow and extends posteriorly to reach the anterior process of the pterygoids (Fig. 1B). The vomer contacts the premaxilla anteriorly, the maxilla anterolaterally, the palatine posterolaterally, and the pterygoid posteriorly (Figs. 1B and 3). The posterior portion of the vomer is relatively short because the anterior process of the pterygoids extends anterior to the level of the foramen palatinum posterius (see Pterygoid below). A similar arrangement is present in Arundelemys dardeni (Evers, Rollot & Joyce, 2021), Dorsetochelys typocardium (Evans & Kemp, 1976), Uluops uluops (Rollot, Evers & Joyce, 2021a), but is absent in Compsemys victa (Lyson & Joyce, 2011) and baenodds (Gaffney, 1972). The shape and contacts of the vomer are similar to those of Arundelemys dardeni (Evers, Rollot & Joyce, 2021). The posterior contact of the vomer with the anterior process of the pterygoids prevents the palatines from contacting their counterparts medially (Fig. 1B). At about mid-length, the vomer bears low, dorsolaterally-directed processes for articulation with the descending process of the prefrontals (Fig. 3A). Similar processes and vomer-prefrontal contacts are found in Arundelemys dardeni (Evers, Rollot & Joyce, 2021), Compsemys victa (Lyson & Joyce, 2011), Glyptops ornatus (Gaffney, 1979), Pleurosternon bullockii (Evans & Kemp, 1975), and Uluops uluops (Rollot, Evers & Joyce, 2021a).

Palatine. The palatine is a laminar bone that forms the posteromedial part of the fossa orbitalis, the posterior margin of the apertura narium interna, the posterior portion of the foramen orbito-nasale, and the anterior and medial margin of the foramen palatinum posterius (Figs. 1B, 3B, and 4D). The lateral aspect of the palatine between the foramen orbito-nasale and the foramen palatinum posterius contacts the maxilla ventrolaterally and contributes minorly to the triturating surface (Fig. 3B). The palatine contacts the descending process of the prefrontal anteriorly, the vomer medially, the pterygoid posteromedially and posteriorly, and the jugal posterolaterally (Figs. 1B, 3, and 4D).

Pterygoid. The anterior half of the pterygoid contacts its counterpart medially, the palatine anterolaterally along a curved and concave suture, the jugal laterally along the anterior portion of the well-developed processus pterygoideus externus, and the vomer anteriorly by means of its anterior process (Figs. 1B, 3B, and 5A, 5B). The posterior half of the pterygoid contacts by means of its posterior process the parabasisphenoid medially, the basioccipital and exoccipital posteromedially, and the quadrate laterally (Figs. 1B, 1D). In the trigeminal region, the pterygoid contacts the purported epipterygoid anterodorsally, the prootic posterodorsally, and the quadrate posteriorly (Fig. 3A). A contact with the descending process of the parietal may have been present but the crushing of the latter does not allow clear observation of that area. The pterygoid forms the posterolateral margin of the foramen palatinum posterius, the ventral margin of the canalis cavernosus, and floors the cavum acustico-jugulare (Figs. 1B and 5B, 5C).

The pterygoid bears a long anterior process that reaches the posteriormost portion of the vomer, and prevents the palatines from contacting each other (Figs. 1B and 3B). This process ends slightly anterior to the level of the anteriormost margin of the foramen palatinum posterius and is about half as long as the palatine anteroposteriorly (Fig. 1B). Such an extensive anterior process is also found in Arundelemys dardeni (Evers, Rollot & Joyce, 2021), Lakotemys australodakotensis (Rollot et al., 2022), Dorsetochelys typocardium (Evans & Kemp, 1976), and Uluops uluops (Rollot, Evers & Joyce, 2021a), and appears to differ from other known paracryptodires (Gaffney, 1972, 1979; Evers, Rollot & Joyce, 2020; Lyson & Joyce, 2011). In some palatobaenines, the pterygoid-palatine suture is located slightly more anteriorly than in other baenodds and the pterygoid marginally protrudes anteriorly within the palatine (Gaffney, 1972; Archibald & Hutchison, 1979; Lyson & Joyce, 2009a). This process is however greatly reduced in comparison to the anterior process of Trinitichelys hiatti. The pterygoid of Trinitichelys hiatti forms a well-developed processus pterygoideus externus that extends into the subtemporal fossa (Figs. 3A and 5). The process is dorsoventrally expanded into a robust, vertical flange that is similar to that of Arundelemys dardeni (Evers, Rollot & Joyce, 2021), Lakotemys australodakotensis (Rollot et al., 2022), Glyptops ornatus (Gaffney, 1979), Pleurosternon bullockii (Evers, Rollot & Joyce, 2020), Pleurosternon moncayensis (Pérez-García et al., 2021), and Uluops uluops (Rollot, Evers & Joyce, 2021a), but also the processus trochlearis pterygoidei of pleurodires. The crista pterygoidei is low and forms the lateral border for the sulcus cavernosus (Fig. 5C). The pterygoid-epipterygoid contact, if real, is inferred to be located along the anterior part of the crista pterygoidei. The canalis cavernosus could not be reconstructed because of the crushing that affects the relevant region of the skull, but the pterygoid clearly exhibits a groove along its dorsal surface that extends posterolaterally for most of its length (Figs. 5B, 5C and 6A, 6B). The anterior portion of this groove is the sulcus cavernosus and the posterior portion corresponds to the ventral margin of the canalis cavernosus. The foramen cavernosum is likely located slightly posteroventral to the anterior limit of the prootic and the pterygoid ventrally contributes to the formation of that foramen. We are not able to determine the exact bony contributions to the foramen nervi trigemini because of the dorsoventral crushing of the skull, but the 3D models suggest that the pterygoid and parietal likely contributed to it (Fig. 3A).

At mid-length along the parabasisphenoid-pterygoid suture, the pterygoid forms the lateral part of a carotid pit (sensu Evers, Rollot & Joyce, 2021; Figs. 5A and 6C), which is roughly half the size of that of Uluops uluops (Rollot, Evers & Joyce, 2021a). The carotid pit of Trinitichelys hiatti is posteriorly constricted by extensions of the pterygoid and parabasisphenoid that partially cover the posterior aspect of the pit and separate it from a narrow, posteriorly directed carotid groove (Figs. 6C, 6D). In that regard, Trinitichelys hiatti differs from Uluops uluops, where the carotid pit is fully confluent posteriorly with the carotid groove and lacks the extensions of the pterygoid and parabasisphenoid that constrict the posterior margin of the carotid pit (Rollot, Evers & Joyce, 2021a). The basicranium of Trinitichelys hiatti thus might exhibit an intermediate ossification stage between that of Uluops uluops and baenodds, in which a foramen posterius canalis carotici interni and canalis caroticus internus are present (see Discussion below). The carotid pit of Trinitichelys hiatti contains two foramina that we identify as the foramen posterius canalis nervi vidiani and the foramen posterius canalis carotici basisphenoidalis (Figs. 5D and 6D). The foramen posterius canalis nervi vidiani is located along the anterolateral margin of the carotid pit and is formed by the pterygoid (Figs. 5D and 6D). The foramen posterius canalis nervi vidiani leads to the canalis nervus vidianus, which extends anteriorly through the pterygoid (Figs. 6B, 6D). The canalis nervus vidianus exits the skull along the dorsal surface of the pterygoid by means of the foramen anterius canalis nervi vidiani, which is located posterolaterally to the anterior process of the pterygoid (Figs. 6B and 6D). The second foramen found in the carotid pit is the foramen posterius canalis carotici basisphenoidalis, through which the cerebral artery enters the parabasisphenoid (Fig. 6D). The foramen posterius canalis carotici basisphenoidalis is located along the medial margin of the carotid pit and formed by the pterygoid laterally and the parabasisphenoid medially. There is no evidence for the presence of a canal for the palatine artery, the canalis caroticus lateralis. When present, this canal generally extends anteriorly along the pterygoid-parabasisphenoid suture and exits the skull through the foramen anterius canalis carotici lateralis (see Rollot, Evers & Joyce, 2021b), usually located within the sulcus cavernosus. In Uluops uluops and Lakotemys australodakotensis, the only known paracryptodires for which a canalis caroticus lateralis can unambiguously be identified, the entrance point of the palatine artery into the skull, the foramen posterius canalis carotici lateralis, is located between the foramen posterius canalis nervi vidiani and the foramen posterius canalis carotici basisphenoidalis (Rollot, Evers & Joyce, 2021a; Rollot et al., 2022). In Trinitichelys hiatti, we are not able to identify any additional foramen in the carotid pit than the posterior foramina for the canals for the vidian nerve and cerebral artery and cannot observe any canal that extends along the pterygoid-parabasisphenoid suture. We consider the canalis caroticus lateralis and the palatine artery to be absent in Trinitichelys hiatti. The circulation system of Trinitichelys hiatti therefore demonstrably resembles that of Arundelemys dardeni (Lipka et al., 2006; Evers, Rollot & Joyce, 2021), Eubaena cephalica (Rollot, Lyson & Joyce, 2018), and Pleurosternon bullockii (Evers, Rollot & Joyce, 2020), taxa in which the canalis caroticus lateralis has been shown to be absent by the use of micro-CT scans. The foramen distalis nervi vidiani is located along the lateral margin of the carotid sulcus just posterior to the extensions of the pterygoid and parabasisphenoid that constrict the carotid pit (Fig. 5D). The foramen distalis nervi vidiani allows the passage of the vidian nerve from the canalis pro ramo nervi vidiani into the carotid sulcus (Fig. 6D).

The pterygoid fossa on the ventral surface of the posterior process of the pterygoid is moderately deep (Fig. 5A). The posterior process of the pterygoid is long and has an elongate contact medially with the basioccipital (Figs. 1B, 5B, and 5D), similar to that of baenodds (Gaffney, 1972; Brinkman & Nicholls, 1991, 1993; Joyce & Lyson, 2015), Arundelemys dardeni (Lipka et al., 2006), and Lakotemys australodakotensis (Rollot et al., 2022), but which contrasts with that of Dorsetochelys typocardium (Evans & Kemp, 1976), Glyptops ornatus (Gaffney, 1979), Pleurosternon bullockii (Evans & Kemp, 1975), Pleurosternon moncayensis (Pérez-García et al., 2021), and Uluops uluops (Rollot, Evers & Joyce, 2021a), in which the posterior process of the pterygoid does not extend beyond the posterior limit of the parabasisphenoid. The posterior process of the pterygoid also contacts the exoccipital posterodorsally as in Arundelemys dardeni (Evers, Rollot & Joyce, 2021) and Saxochelys gilberti (Lyson, Sayler & Joyce, 2019). The ventral surface of the posterior process of the pterygoid is smooth as anterior tubercula basioccipitale are absent (Figs. 5A and 6C). Within the cavum acustico-jugulare, the pterygoid contacts the prootic anterodorsomedially and the quadrate laterally. A contact with the opisthotic is not preserved but was likely present dorsomedially.

Epipterygoid. The area of the descending process of the parietal is badly crushed (Fig. 4A). Numerous fragmentary bony pieces can be found in the matrix surrounding this region of the skull and an attribution of these to either the parietal, pterygoid, or a potential epipterygoid is not possible with certainty. We are unable to determine if a separate, ossified epipterygoid is present in Trinitichelys hiatti, but note that an intriguing piece of bone, located medial to the processus pterygoideus externus, may represent a portion of an epipterygoid (Fig. 3A). Its bony components have been segmented and are described herein. The piece of bone of interest is laterally thick and parasagittaly crossed by a suture that is visible along the lateral surface where the bony element is the thickest. The thickening and position of the suture is similar to the epipterygoid of Uluops uluops and its suture with the parietal. The epipterygoid of Uluops uluops has a lateral bulge at its dorsal process, which extends as a thick ridge over the lateral surface of the epipterygoid, and the suture with the parietal is located just dorsal to that (Rollot, Evers & Joyce, 2021a). If the conformation is the same in Trinitichelys hiatti, then the bone located dorsomedially to the suture within the bony element belongs to the parietal, and the ventrolateral bone corresponds to the epipterygoid. An alternative interpretation is that the bone medial to the suture is actually part of an expanded crista pterygoidea, and that the bone lateral to the suture corresponds to the ventralmost aspect of the descending process of the parietal that laterally rests on the pterygoid. As we are not able to favor one hypothesis over the other, we consider the presence or absence of an epipterygoid in Trinitichelys hiatti to be unknown.

Quadrate. The quadrate forms most of the cavum tympani and antrum postoticum, the condylus mandibularis, the lateral portion of the cavum acustico-jugulare, and the incisura columella auris (Figs. 1D, 1E and 3A). In lateral view, the quadrate contacts the quadratojugal along a convex suture and the squamosal posterodorsally (Figs. 1E and 3A). The cavum tympani is deep and visually separated from the voluminous antrum postoticum by a ridge formed by the quadrate. Within the lower temporal fossa, the quadrate has an elongate contact with the pterygoid medially. Within the upper temporal fossa, the quadrate contacts the prootic anteromedially, the opisthotic posteromedially, and the squamosal posterolaterally (Figs. 1A and 3A). The presence of a potential contact between the supraoccipital and quadrate is unclear. A small piece of bone is located between the supraoccipital, prootic, opisthotic, and quadrate on the floor of the upper temporal fossa, but we are unable to determine with confidence which element it represents (Fig. 1A, right side). Its attribution to the supraoccipital or quadrate would suggest a point contact between these two bones, attribution to the opisthotic or prootic would preclude such a contact, even though both bones would still approach one another closely. A supraoccipital-quadrate contact is commonly present in paracryptodires, varying from an elongate contact as in Compsemys victa (Lyson & Joyce, 2011) and Eubaena cephalica (Rollot, Lyson & Joyce, 2018), to a point contact, as in Chisternon undatum (Gaffney, 1972) and, likely, Pleurosternon bullockii (Evers, Rollot & Joyce, 2020). The presence of a supraoccipital-quadrate contact in Trinitichelys hiatti, even if reduced, would thus correspond to the usual condition found in paracryptodires. The quadrate forms the lateral margin of the foramen stapedio-temporale, canalis stapedio-temporalis, and aditus canalis stapedio-temporalis, and the posterodorsal margin of the canalis cavernosus and the associated posterior foramen. Two facets separated by a shallow sulcus are present on the left condylus mandibularis for the articulation with the mandible (Fig. 1B). The quadrate forms the lateral half of the processus trochlearis oticum, which seems somewhat more prominent than in Arundelemys dardeni (Evers, Rollot & Joyce, 2021). Within the cavum acustico-jugulare, the quadrate contacts the opisthotic dorsomedially and the pterygoid ventromedially.

Prootic. The prootic is badly crushed on both sides of the skull. We are able to determine the contacts of the prootic with surrounding bones, but its contributions to internal structures can only be estimated. The prootic contacts the parietal anteromedially, the pterygoid anteroventrally, the quadrate laterally and posterolaterally, the supraoccipital posteromedially, and the opisthotic posteriorly below the skull surface (Figs. 1A and 3A). A point contact with the opisthotic is perhaps present within the upper temporal fossa (see Quadrate above). A contribution of the prootic to the foramen nervi trigemini remains unclear as this area is damaged (Fig. 3A). Within the upper temporal fossa, the prootic forms the medial margin of the foramen stapedio-temporale (Fig. 1A), which leads to the cavum acustico-jugulare by means of the canalis stapedio-temporalis and aditus canalis stapedio-temporalis that are medially bordered by the prootic as well. Within the cavum acustico-jugulare, the prootic borders the canalis cavernosus dorsomedially. The prootic also forms the dorsal margin of the foramen cavernosum. The prootic contributes to the anterolateral part of the cavum cranii, but dorsoventral crushing obscures the fossa acustico-facialis and the canals for the facial (VII) and acoustic nerves (VIII). The canals for the vidian nerve, which are located in the pterygoid (see above), however, provide clues about the path of the facial nerve and the relative position of the geniculate ganglion. The vidian nerve extends ventromedially from the medial part of the pterygoid that housed the lateral head vein to the carotid groove ventrally (Figs. 5D, 6B, and 6D), meaning that the geniculate ganglion was likely located within the canalis cavernosus, and that the facial nerve, which is usually medial to the ganglion, extended laterally from the fossa acustico-facialis to the canalis cavernosus. This is also the case in Arundelemys dardeni (Evers, Rollot & Joyce, 2021), Lakotemys australodakotensis (Rollot et al., 2022), Eubaena cephalica (Rollot, Lyson & Joyce, 2018), Pleurosternon bullockii (Evers, Rollot & Joyce, 2020), Pleurosternon moncayensis (Pérez-García et al., 2021), and Uluops uluops (Rollot, Evers & Joyce, 2021a). The prootic otherwise forms the anterior part of the cavum labyrinthicum and the anterior half of the canalis semicircularis anterior and horizontalis. The fenestra ovalis is not well preserved in the specimen, making it impossible to judge if it was completely surrounded by the prootic and opisthotic ventrally.

Opisthotic. The right opisthotic is well-preserved and only shows minor signs of damage that affect its most anterior part and the processus interfenestralis, which is almost completely missing. The anterior part of the opisthotic contacts the supraoccipital anteromedially, the prootic anteriorly below the skull surface, and the quadrate anterolaterally (Fig. 1A). The posterolaterally oriented paroccipital process contacts the exoccipital ventromedially and the squamosal anterolaterally, roofs the cavum acustico-jugulare, and forms the dorsal margin of the fenestra postotica (Figs. 1B, 1D). The paroccipital process bears a ridge along its posterodorsal surface that extends posterolaterally and disappears close to the suture with the squamosal (Fig. 1D). A near-identical ridge is present on the paroccipital process of Lakotemys australodakotensis (Rollot et al., 2022) and Uluops uluops (Rollot, Evers & Joyce, 2021a), but absent in Arundelemys dardeni (Evers, Rollot & Joyce, 2021). The opisthotic of Trinitichelys hiatti does not contribute to the foramen stapedio-temporale, but a small contribution to the canalis stapedio-temporalis cannot be excluded. The opisthotic forms the posterior part of the cavum labyrinthicum, the posterior half of the canalis semicircularis horizontalis and canalis semicircularis posterior, and the foramen externum nervi glossopharyngei (IX) at the base of the broken processus interfenestralis. Due to the abovementioned damage, we are not able to comment on the anatomy of the fenestra perilymphatica, the foramen jugulare anterius, the recessus scalae tympani, the fenestra ovalis, and the foramen internum nervi glossopharyngei.

Supraoccipital. The supraoccipital forms the posterior tip of the skull roof, roofs the cavum cranii, and forms the dorsal margin of the foramen magnum and the medial margin of the upper temporal fossa (Figs. 1A, 1D, and 3B). The supraoccipital contacts the parietal anteromedially, the prootic anterolaterally, the opisthotic posterolaterally, and the exoccipital posteroventrally (Figs. 1A, 1D, and 3). A medial point contact with the quadrate might be present as well (see Quadrate above). We note that in Arundelemys dardeni and Lakotemys australodakotensis, with which Trinitichelys hiatti shares a large amount of features (Evers, Rollot & Joyce, 2021; Rollot et al., 2022; this study) and that were regularly found close to the latter in paracryptodiran phylogenies (Lyson & Joyce, 2011; Pérez-García et al., 2015; Rollot, Evers & Joyce, 2021a), the supraoccipital is prevented from contacting the quadrate by a prootic-opisthotic contact just lateral to the latter (Lipka et al., 2006; Rollot et al., 2022). If the bony piece of interest is part of the prootic in Trinitichelys hiatti, then the bone arrangement within its upper temporal fossa would be nearly identical to that of Arundelemys dardeni and Lakotemys australodakotensis. On the other hand, if the bony piece belongs to the supraoccipital, the latter would then have a long, pointed lateral process that would be unique among paracryptodires, with the exception of one specimen of Palatobaena cohen (DMNH EPV.97017), in which differences within that portion of the skull are attributed to intraspecific or ontogenetic variation (Lyson & Joyce, 2009a). As we are unable to favor one hypothesis over the other and both seem equally admissible to us, we choose to not attribute the small piece of bone in the center of the upper temporal fossa of Trinitichelys hiatti to either the supraoccipital or prootic. The crista supraoccipitalis is a vertical sheet of bone developed between the foramen magnum and the skull roof (Fig. 3B). In dorsal view, the crista supraoccipitalis is short and posteriorly only slightly extends beyond the level of the condylus occipitalis, but does not reach the level of the posterior tip of the squamosals (Fig. 1A). The crista supraoccipitalis is mediolaterally thin and the ventrodorsal depth of the upper temporal fossa is intermediate (Fig. 1D), as in Pleurosternon bullockii (Evers, Rollot & Joyce, 2020). On the skull roof, the parietals cover the crista supraoccipitalis almost completely. The posterodorsal tip of the crista supraoccipitalis is apparent just posterior to the parietals on the skull roof and forms the posteromedial end of the latter (Fig. 1A). The contribution of the crista supraoccipitalis to the skull roof is, however, extremely minor, and is similar to the condition observed in Cedrobaena putorius (Lyson & Joyce, 2009b), Neurankylus eximius (Brinkman & Nicholls, 1993), and Pleurosternon bullockii (Evers, Rollot & Joyce, 2020), but differs from Dorsetochelys typocardium (Evans & Kemp, 1976), Eubaena cephalica (Rollot, Lyson & Joyce, 2018), and Uluops uluops (Rollot, Evers & Joyce, 2021a), in which the posterodorsal aspect of the crista supraoccipitalis is developed as a large plate. The supraoccipital roofs the cavum labyrinthicum and forms the posterior portion of the canalis semicircularis anterior and the anterior portion of the canalis semicircularis posterior.

Basioccipital. The basioccipital is an unpaired, quadrangular element that forms the posteroventromedial portion of the skull, floors the posterior part of the cavum cranii and forms the ventral margin of the foramen magnum (Figs. 1B, 1D, and 3B). In posterior view, the basioccipital contacts the exoccipital dorsolaterally and the pterygoid laterally and forms the complete articular surface of the condylus occipitalis (Fig. 1D), as in Arundelemys dardeni (Evers, Rollot & Joyce, 2021), Glyptops ornatus (Gaffney, 1979), but also Compsemys victa, Dorsetochelys typocardium, Kallokibotion bajazidi, and Uluops uluops (Rollot, Evers & Joyce, 2021a), but likely not baenodds (Rollot, Evers & Joyce, 2021a). The basioccipital bears a low crista dorsalis basioccipitalis along its anterodorsal surface and laterally forms short tubercula basioccipitale to which the exoccipital and pterygoid contribute as well (Fig. 1D). In ventral view, the basioccipital contacts the parabasisphenoid anteriorly and the pterygoid laterally along nearly straight sutures (Fig. 1B). The ventral surface of the basioccipital is nearly flat and only a weak depression is apparent posterior to the contact with the parabasisphenoid. We are not able to identify a canalis basioccipitalis within that depression. The basioccipital likely contacted the processus interfenestralis of the opisthotic along its anterodorsal surface, but the exact extent of this contact could not be assessed as the process is not preserved.

Exoccipital. The exoccipital in Trinitichelys hiatti is not fused to the basioccipital and remains a separate bony element (Figs. 1D and 3B), contra most baenodds (Gaffney, 1982). The exoccipital forms the posterolateral part of the cavum cranii, the lateral margin of the foramen magnum, the posteromedial part of the cavum acustico-jugulare, the medial margin of the fenestra postotica, and the posterior margin of the foramen jugulare anterius (Figs. 1D and 3B). In posterior view, the exoccipital contacts the supraoccipital dorsomedially, the paroccipital process of the opisthotic dorsally and dorsolaterally, the pterygoid ventrolaterally, and the basioccipital ventrally (Fig. 1D). The exoccipital contributes to the tuberculum basioccipitale dorsolaterally (Fig. 1D), which is typical in baenodds (Gaffney, 1982), but different from Neurankylus torrejonensis, where they are formed by the basioccipital only (Lyson et al., 2016). The exoccipital does not contribute to the formation of the functional articular surface of the condylus occipitalis (see Basioccipital above). Within the cavum acustico-jugulare, the exoccipital contacts the opisthotic anteriorly, the pterygoid anterolaterally and laterally, and rests on the dorsal surface of the basioccipital. We are not able to determine the number of foramina nervi hypoglossi because of the dorsoventral crushing of that area, but note the presence of at least one canal for the hypoglossal nerve that crosses the left exoccipital mediolaterally.

Parabasisphenoid. The parabasisphenoid is a single element that forms the ventral margin of the cavum cranii (Fig. 3B). The parabasisphenoid contacts the pterygoid laterally along its entire length, the prootic dorsolaterally posterior to the level of the foramen nervi trigemini, and the basioccipital posteriorly (Figs. 1B and 3B). A contact with the palatine and vomer is absent (Fig. 1B). The anterior half of the parabasisphenoid, which forms the rostrum basisphenoidale, is developed as a thin sheet of bone that rests on the pterygoid for all of its length, forms the medial margin of the sulcus cavernosus, gradually broadens towards the posterior, and is bordered by the sella turcica posteriorly and the retractor bulbi pits dorsolaterally (Fig. 6A). The sella turcica forms a moderately deep depression that houses the two foramina anterius canalis carotici basisphenoidalis, which are relatively widely spaced, and is overhung by the dorsum sellae (Fig. 6B). Most of the clinoid processes is missing, but their bases are preserved and show that those are broad and that the clinoid processes likely extended anterolaterally to the dorsum sellae. The retractor bulbi pits of Trinitichelys hiatti are located ventrally to the base of the clinoid processes and form shallow depressions, as in Arundelemys dardeni (Evers, Rollot & Joyce, 2021), that contrast with the great depth of the retractor bulbi pits observed in Pleurosternon bullockii (Evers, Rollot & Joyce, 2020) and Uluops uluops (Rollot, Evers & Joyce, 2021a). The posterior part of the parabasisphenoid is dorsoventrally thick and its dorsal surface forms a deep concavity that contained parts of the brain. The foramen posterius canalis nervi abducentis is located along the anterolateral aspect of the dorsal surface of the parabasisphenoid (Fig. 6A). The short canalis nervus abducentis extends anteriorly through the parabasisphenoid and exits the latter by means of the foramen anterius canalis nervi abducentis, which is located within the retractor bulbi pits (Fig. 6B). The foramen anterius canalis nervi abducentis, canalis nervus abducentis, and foramen posterius canalis nervi abducentis are solely formed by the parabasisphenoid (Figs. 6A, 6B). A contribution of the pterygoid to the formation of these structures, as observed in Arundelemys dardeni (Evers, Rollot & Joyce, 2021), Pleurosternon bullockii (Evers, Rollot & Joyce, 2020), and Uluops uluops (Rollot, Evers & Joyce, 2021a), is absent. The lateral margins of the posterior part of the parabasisphenoid form raised surfaces for the articulation with the prootic and likely bordered the hiatus acusticus ventrally. A crista basis tuberculi basalis is not fully preserved in Trinitichelys hiatti, but a small portion of the parabasisphenoid that is slightly raised posteromedially just anterior to the suture with the basioccipital indicates that at least a short crista basis tuberculi basalis was present, as in Arundelemys dardeni (Evers, Rollot & Joyce, 2021) and Lakotemys australodakotensis (Rollot et al., 2022).

In ventral view, the parabasisphenoid forms the medial margin of the small carotid pit at mid-length along the suture with the pterygoid (Figs. 1B and 6C). The foramen posterius canalis nervi vidiani and foramen posterius canalis carotici basisphenoidalis are present within this cavity (see Pterygoid above). The foramen posterius canalis carotici basisphenoidalis is located along the medial margin of the carotid pit and formed by the pterygoid laterally and the parabasisphenoid medially (Fig. 6D). As mentioned in the pterygoid section, no canal other than the canalis nervus vidianus and canalis caroticus basisphenoidalis extends through the basicranium. The palatine artery and canalis caroticus lateralis are therefore considered absent, as is the case in Arundelemys dardeni (Lipka et al., 2006), Pleurosternon bullockii (Evers, Rollot & Joyce, 2020), Eubaena cephalica (Rollot, Lyson & Joyce, 2018), and likely all baenodds (Rollot, Evers & Joyce, 2021a). In Pleurosternon bullockii (Evers, Rollot & Joyce, 2020) and Uluops uluops (Rollot, Evers & Joyce, 2021a), the parabasisphenoid bears a basipterygoid process that extends laterally into the pterygoid and roofs the carotid pit for its entire length. In Trinitichelys hiatti, the dorsal aspect of the carotid pit is fully formed by the pterygoid and we are not able to observe a lateral process of the parabasisphenoid in that area of the skull (Figs. 4A and 5C). The basipterygoid process is thus considered absent in this taxon, as is also the case in Arundelemys dardeni (Evers, Rollot & Joyce, 2021), Lakotemys australodakotensis (Rollot et al., 2022), and advanced baenids (Gaffney, 1982). Posterior to the carotid pit, the parabasisphenoid forms along with the pterygoid a groove that extends posteriorly on the ventral surface of the skull and that likely housed the internal carotid artery (Fig. 6C). The posterolateral surface of the parabasisphenoid is smooth and anterior tubercula basioccipitale are absent (Figs. 1B and 6C).

Stapes. The right stapes is partially preserved and shattered into four pieces that do not preserve its more medial and lateral parts. The four pieces are oval shaped in cross-section, have a similar width and show no sign of expansion, suggesting that they belong to the central aspect of the stapes.

Hyoid apparatus. Portions of the first pair of ceratobranchials are preserved on the underside of the skull of Trinitichelys hiatti. The right ceratobranchial is complete (see Fig. 7) while only the posterior part is preserved for the left one. Both ceratobranchials appear to be slightly shifted towards the right. Ceratobranchial I is elongate and curved. Its anterior portion extends to the level of the processus pterygoideus externus and its posterior portion reaches the level of the posterior margin of the opisthotic. The ceratobranchial I is overall circular in cross-section posteriorly and broadens progressively towards the anterior, to become dorsoventrally broader along its anterior half. The anterior and posterior articulation facets are not preserved.

Discussion

Comparison with the original description of Gaffney (1972)

Trinitichelys hiatti was originally named and figured by Gaffney (1972) based on a partial skeleton, which only lacks the lower jaw, the caudal vertebrae, the distal parts of the limbs, and the posterior part of the shell (Gaffney, 1972; Joyce & Lyson, 2015). Gaffney (1972) provided illustrations of the skull that showed some bony contacts in dorsal and left lateral views, but did not provide a description beyond the characters listed in his diagnosis. A detailed documentation of the skull was, therefore, lacking prior to the present study.

We were able to identify most cranial sutures from the available µCT slice data, allowing us to highlight in detail all bony contacts with exception of the one between the jugal and quadratojugal. Our reconstruction shows several stark differences with the original one of Gaffney (1972). The sutures on the skull surface are extremely difficult, if not impossible, to interpret directly on the specimen, and we were only able to trace them with confidence by using µCT scans and digitally reconstructing the skull bone by bone. We suspect that most of the sutures identified by Gaffney actually correspond to cracks, of which a multitude cross the skull. This explains the differences between his original interpretation and our reconstruction of the skull. We here take the opportunity to review these differences based on the original restored dorsal, ventral, and lateral views provided by Gaffney (1972; Fig. 8).

Figure 8 Comparison of the interpretations of the cranial sutures of the skull of Trinitichelys hiatti (MCZ VPRA-4070, holotype) between Gaffney (1972) and our study.

(A) Dorsal view of the skull, (B) original illustration in dorsal view taken from Gaffney (1972), (C) illustration in dorsal view following our interpretation of the cranial sutures, (D) ventral view, (E) original illustration in ventral view taken from Gaffney (1972), (F) illustration in ventral view following our interpretation of the cranial sutures, (G) right lateral view of the skull, (H) original illustration in left lateral view (mirrored) taken from Gaffney (1972), and (I) illustration in right lateral view following our interpretation of the cranial sutures. Abbreviations: bo, basioccipital; fr, frontal; j-qj, juga-quadratojugal; ju, jugal; mx, maxilla; na, nasal; op, opisthotic; pa, parietal; pal, palatine; pbs, parabasisphenoid; pf, prefrontal; pmx, premaxilla; po, postorbital; pro, prootic; pt, pterygoid; qj, quadratojugal; qu, quadrate; so, supraoccipital; sq, squamosal; vo, vomer.

In dorsal view, Gaffney (1972) identified a large nasal that slightly contributes to the anterodorsal margin of the orbit and contacts the maxilla laterally and the frontal dorsally (Fig. 8B). Although some uncertainty remains for the presence of a nasal-maxilla contact (see Nasal in the Description section above), the nasal appears to be about half the size of that inferred by Gaffney (1972) and contacts the prefrontal posterolaterally and the frontal posteriorly (Fig. 8C). The posterior extent of the nasal is reduced compared to Gaffney (1972) reconstruction and the nasal is prevented from contributing to the orbital margin by an extended contact between the prefrontal and the maxilla. Gaffney (1972) also interpreted the interorbital space as being fully occupied by the frontals, the latter forming most of the dorsal margin of the orbit, and the dorsal plate of the prefrontals as being reduced and not exposed on the dorsal skull (Fig. 8B). Our reconstruction of that area alternatively shows that the dorsal plate of the prefrontal is actually moderately exposed on the skull roof, forming the anterior half of the dorsal aspect of the orbital margin, and that the frontal only slightly contributes to the latter by means of a short lateral process (Fig. 8C). Gaffney (1972), additionally, inferred a point contact between the parietal and squamosal along the anterolateral part of the upper temporal emargination on the left side of the skull, while on the right side, this contact seems to be prevented by an extremely small contribution of the postorbital to the upper temporal emargination (Fig. 8B). Instead, we identify a contact between the parietal and squamosal on the right side of the skull, albeit small, but note that too much damage affects the same area on the left side to allow a clear observation of the contact there (Fig. 8C).

In ventral view, Gaffney (1972) only showed a maxilla-palatine contact along the medial margin of the triturating surface and an opisthotic-quadrate contact posterior to the incisura columella auris (Fig. 8E), which we are able to confirm. Gaffney (1972) furthermore traced dashed lines posteromedially to the foramina palatinum posterius, indicating the likely position of the pterygoid-pterygoid, pterygoid-palatine, and pterygoid-vomer sutures, with the anterior limit of the pterygoids being located just posterior to the foramina palatinum posterius (Fig. 8E). Our interpretation of that area differs in that we identify an anterior process of the pterygoids that anteriorly protrudes between the palatines and contacts the vomer anterior to the anterior margin of the foramina palatinum posterius (Fig. 8F). The suture between the pterygoid and vomer is, therefore, located more anteriorly than Gaffney (1972) inferred, and the vomer appears to be anteroposteriorly reduced compared to the original interpretation of that specimen.

In lateral view, the main difference lies in the interpretation of the sutures, contacts, and shape of the jugal. Gaffney (1972) identified a rather anteroposteriorly elongate jugal that contributes to the orbital margin and contacts the maxilla anteroventrally, the postorbital dorsally, and the quadratojugal posteriorly (Fig. 8H). Instead, we suggest that the jugal is located slightly more posterodorsal to what Gaffney (1972) interpreted, and is prevented from contributing to the orbit by a short contact between the postorbital and maxilla along the posteroventral corner of the orbit (Fig. 8I). We are also unable to identify a suture between the jugal and quadratojugal, and suggest that the feature identified by Gaffney (1972) actually corresponds to a crack. Although those differences and new interpretations do not considerably affect the previously suggested affinities and phylogenetic position of Trinitichelys hiatti (see Phylogenetic relationships below), our description and interpretations provide new insights into the anatomy of that taxon, allowing us to compare it in detail to other known basal baenids and highlight intriguing similarities shared by basal baenids with both pleurosternids and more advanced baenids (see Similarities with pleurosternids and advanced baenids section below).

Validity of Arundelemys dardeni, Lakotemys australodakotensis, and Trinitichelys hiatti

The reinterpretation of the cranial anatomy of Trinitichelys hiatti raises questions regarding the validity of the subsequently named Early Cretaceous Arundelemys dardeni and Lakotemys australodakotensis, as these two taxa were named presuming Trinitichelys hiatti to have a morphology that indeed is very different, but herein shown to be incorrect. While several features clearly distinguish Lakotemys australodakotensis from both Arundelemys dardeni and Trinitichelys hiatti, such as the presence of a canalis caroticus lateralis, the absence of a lateral process of the frontal, the ventral exposure of the parabasisphenoid for most of its length, the formation of the foramen palatinum posterius by the palatine, and its older occurrence relative to the other two taxa (Berriasian to Valanginian), the reconstructions of Trinitichelys hiatti and Arundelemys dardeni show greater similarity that warrant further exploration. We therefore thoroughly compare and discuss the differences and variation between these species based on available 3D segmented skull models (Evers, Rollot & Joyce, 2021; this study), and provide a rationale for retaining both as valid taxa.

While Lakotemys australodakotensis can easily be distinguished from Arundelemys dardeni and Trinitichelys hiatti, the differences between the latter two are more subtle and we are not able to identify clear differences in terms of bony contacts or bony contributions to most cranial structures. We are nonetheless able to identify a multitude of more subtle differences that generally pertain to the size and shape of several bones or bony structures: Trinitichelys hiatti has a smaller orbit, a greater height of the maxilla below the orbit, smaller nasals that contact one another for most of their length, a greater exposure of the prefrontal on the skull roof, a foramen praepalatinum located closer to the premaxilla-vomer suture, a protruding ridge formed by the nasals ventromedially that divides the nasal cavity into left and right halves, a taller premaxilla, more transversely oriented canalis caroticus basisphenoidalis, a medial extension of the descending process of the prefrontals within the nasal cavity but sheet-like ridge absent, a more prominent processus trochlearis oticum, a transverse ridge along the posterodorsal surface of the paroccipital process of the opisthotic, a poorly developed mediolateral ridge of the parietal (which forms the posterior margin of the roof of the sulcus palatino-pterygoideus), absence of a pterygoid contribution to the foramen anterius canalis nervi abducentis and canalis nervus abducentis (which are completely formed by the parabasisphenoid), a slightly more reduced ventral exposure of the parabasisphenoid, and a narrower incisura columella auris. Additionally, the frontal of Trinitichelys hiatti has a straight suture anteriorly with the nasal, whereas the frontal clearly protrudes anteriorly between the nasals in Arundelemys dardeni (Evers, Rollot & Joyce, 2021), separating the latter along the midline for half of their length. Some of these differences pertain to features known to vary intraspecifically in extant turtles (the trionychid Apalone ferox: Dalrymple, 1977; the emydid Pseudemys texana: Bever, 2009a, the kinosternid Sternotherus odoratus: Bever, 2009b) or to features that have been interpreted as such for fossils (the protostegid Rhinochelys pulchriceps: Evers, Barrett & Benson, 2019). Some elements of discussion have also been provided for baenids, such as Palatobaena cohen (Lyson & Joyce, 2009a) and Saxochelys gilberti (Lyson, Sayler & Joyce, 2019), but only the variable ventral exposure of the parabasisphenoid that was observed in specimens of Eubaena cephalica is relevant to us here (Rollot, Lyson & Joyce, 2018). We, therefore, base the following comparisons on these studies.

As the holotype skull of both species are nearly identical in size (a 58 mm full skull length for Arundelemys dardeni likely from the premaxilla to the posterior margin of what is preserved from the skull, slightly posterior to the base of the condylus occipitalis according to Lipka et al., 2006; a 56 mm premaxilla-condylus occipitalis length for Trinitichelys hiatti according to Gaffney, 1972), we can dismiss the possibility that variation between them results from ontogenetic changes. A ventromedially protruding ridge that separates the nasal cavity into left and right halves, as observed in Trinitichelys hiatti but not in Arundelemys dardeni, is variably present or absent in the Pseudemys texana specimens used by Bever (2009a), and this feature therefore appears to be a poor taxonomic delineator. This is maybe unsurprising, as the ridge likely serves as the attachment for an internarial septum, and soft tissue attachment sites are known to frequently vary.

The degree of closure of the fissura ethmoidalis has been shown to exhibit some variability as well, as it can be closed, partially closed, or narrowly open, and has its ventral margin that continues to ossify late in postnatal ontogeny in Pseudemys texana (Bever, 2009a). In Arundelemys dardeni and Trinitichelys hiatti, the medial extension of the descending process of the prefrontal frames the fissura ethmoidalis. The fissura ethmoidalis is narrower in Trinitichelys hiatti, which might be due to the presence of a sheet-like ridge on the medial extension of the descending process, reminding of the differences observed between specimens of Pseudemys texana (Bever, 2009a).

On the skull roof, the nasals of Arundelemys dardeni are larger than that of Trinitichelys hiatti and they differ in shape, and the frontal of Arundelemys dardeni clearly protrudes anteriorly to separate the nasals along the midline for half of their length, while the frontal of Trinitichelys hiatti has an anterior straight suture with the nasals. The anterior process of the frontal, including its anterior apex, have been shown to exhibit a great degree of variability in size and shape in specimens of Pseudemys texana (Bever, 2009a), leading others to also consider great nasal variation to be of limited taxonomic importance in a set of specimens of the fossil Rhinochelys pulchriceps (Evers, Barrett & Benson, 2019).

On the otic capsule, the processus trochlearis oticum appears to be more prominent in Trinitichelys hiatti than in Arundelemys dardeni, a difference that could potentially be a result of sexual dimorphism, as is the case in Pseudemys texana in which the processus trochlearis oticum is more distinctively sculptured in adult males than females (Bever, 2009a), but we are not able to determine sexual dimorphism herein given that we only have two skulls for study and no particular way to determine their sex. In Apalone ferox, Dalrymple (1977) similarly noticed differences in the size of the processus trochlearis oticum, but suggested that these differences, along with variation of other cranial features such as the size and shape of triturating surfaces, height and width of the supraoccipital crest, and width of the temporal passageway, are actually linked to the feeding mechanism.

Despite damage to the quadrate in Arundelemys dardeni, the incisura columella auris appears to be intact and is broader than that of Trinitichelys hiatti. Variation to this feature has been reported in Sternotherus odoratus, in which the incisura columella auris can be open, restricted caudally by a descending secondary process of the quadrate, or restricted by an ascending secondary process of the quadrate (Bever, 2009b). The foramen anterius canalis nervi abducentis and anteriormost portion of the canalis nervus abducentis are completely formed by the parabasisphenoid in Trinitichelys hiatti, while a pterygoid contribution to these structures is present in Arundelemys dardeni. The amount of variation for this character is still poorly known in paracryptodires, but in Pseudemys texana, in which the foramen anterius canalis nervi abducentis and canalis nervus abducentis are generally completely formed by the parabasisphenoid, at least one specimen has been observed to exhibit a pterygoid contribution to the latter structures (Bever, 2009a). Finally, in the baenid Eubaena cephalica, the parabasisphenoid of a newly discovered specimen has been shown to be slightly more elongate than that of the holotype (Rollot, Lyson & Joyce, 2018), a very similar difference to that observed between Arundelemys dardeni and Trinitichelys hiatti.

Among the 15 anatomical differences we observed between Arundelemys dardeni and Trinitichelys hiatti, eight have previously been described as variable features in some other turtles. The remaining seven differences have not been identified as being subject to variation in this selection of turtles, which does not necessarily mean that other populations of the same species or other species of turtles could not actually show variability in some or all of these aspects of cranial morphology. Unfortunately, studies of cranial variation (postnatal ontogeny, sexual dimorphism, individual) remain extremely rare in turtles and assessing whether differences as observed herein between two fossil skulls are part of intraspecific variation or may actually represent interspecific variation is extremely challenging. More studies such as those on Pseudemys texana (Bever, 2009a) and Sternotherus odoratus (Bever, 2009b) are needed to better understand cranial morphological variability in turtles. While waiting for more insights into this subject, the numerous differences identified herein, albeit minor, are sufficient for now to keep both species as separate taxonomic entities, but we are aware that future studies on cranial morphological variability or additional fossil discoveries of material of Arundelemys dardeni and Trinitichelys hiatti might provide new arguments rather in favor of a synonymy.

Finally, obtaining more precise data about the age of Arundelemys dardeni and Trinitichelys hiatti could help solving this issue. Arundelemys dardeni and Trinitichelys hiatti were found in Aptian-Albian layers of Maryland and Texas, respectively, but a more constrained age is not known yet, which impedes synonymizing or clearly distinguishing both taxa based on temporal considerations. Arundelemys dardeni was originally described as being late Albian to early Aptian in age (Lipka et al., 2006) but this was likely a typographical error, as a late Aptian to early Albian age was more recently mentioned for the layers in which the skull had been found (Frederickson, Lipka & Cifelli, 2018). Trinitichelys hiatti was first described as Albian in age by Gaffney (1972), although he also mentioned a slightly more precise age for the latter as being early Albian, based on Stephenson et al. (1942). Recent work, however, on the stratigraphy of Early Cretaceous formations in Arkansas, Oklahoma, and Texas, rather suggest a global Aptian-Albian age for the Antlers Formation, in which Trinitichelys hiatti was found (for additional geological information and summaries, see Cifelli et al., 1997; Nydam & Cifelli, 2002; Suarez et al., 2021). Given that the Aptian and Albian span over a time period of about 30 Mya and a restricted occurrence time is lacking for both taxa, synonymizing them appears to be out of the question until detailed stratigraphical data allow a determination of a precise age for Arundelemys dardeni and Trinitichelys hiatti.

Similarities with pleurosternids and advanced baenids

The basal baenids Arundelemys dardeni and Lakotemys australodakotensis have recently been shown to share an intriguing combination of features either typical of pleurosternids or of baenids (Rollot et al., 2022). The redescription of Trinitichelys hiatti we provide herein allows extending these preliminary comparisons to the latter, and all basal baenid skulls known to date share this combination of features. The features of interest are summarized in Table 1 and we briefly discuss them here again. Arundelemys dardeni, Lakotemys australodakotensis, and Trinitichelys hiatti notably share with pleurosternids an anterior process of the pterygoid, a well-developed external process of the pterygoids, an ossified epipterygoid, relatively narrow triturating surfaces, and a moderate exposure of the prefrontal on the skull roof. Typical features of more advanced baenids that are shared by Arundelemys dardeni, Lakotemys australodakotensis, and Trinitichelys hiatti but not with pleurosternids are the presence of a posterior process of the pterygoid with an elongate contact with the basioccipital (reduced contact in pleurosternids), the absence of secondary basioccipital tubera formed by the parabasisphenoid along the posterior portion of its suture with the pterygoid (present in pleurosternids), and the absence of a basipterygoid process (present in pleurosternids). Given this combination of features, Arundelemys dardeni and Lakotemys australodakotensis were suggested to be morphological intermediates between pleurosternids and more advanced baenids (Rollot et al., 2022) and the fact that Trinitichelys hiatti shares with the latter two taxa the same similarities reinforces this hypothesis, which is discussed later in this paper (see Phylogenetic relationships below).

Table 1 Comparative table summarizing the combination of features shared by Trinitichelys hiatti, Arundelemys dardeni and Lakotemys australodakotensis, typical of either Pleurosternidae or Baenidae.

Character	Pleurosternon bullockii	Uluops uluops	Arundelemys dardeni	Lakotemys australodakotensis	Trinitichelys hiatti	Eubaena cephalica	Palatobaena knellerorum	
Anterior process of the pterygoid	Present	Present	Present	Present	Present	Absent	Absent	
External process of the pterygoid	Well-developed	Well-developed	Well-developed	Well-developed	Well-developed	Reduced	Reduced	
Ossified epipterygoid	Present	Present	Present	Present	?	Absent	Absent	
Triturating surfaces	Narrow	Narrow	Narrow	Narrow	Narrow	More expanded	More expanded	
Exposure of the prefrontal on skull roof	Moderate	Moderate	Moderate	Moderate	Moderate	Absent	Reduced	
Pterygoid/basioccipital contact	Reduced	Reduced	Elongate	Elongate	Elongate	Elongate	Elongate	
Anterior tubercula basioccipitale	Present	Present	Absent	Absent	Absent	Absent	Absent	
Basipterygoid process	Present	Present	Absent	Absent	Absent	Absent	Absent	

Circulatory system

The circulatory system has been extensively used in turtle systematics for decades and the position of the foramen posterius canalis carotici interni was used to distinguish the primary groups of turtles from one another (McDowell, 1961; Gaffney, 1975; Sterli & de la Fuente, 2010). Different interpretations for the circulatory system of paracryptodires, however, have recently been proposed, especially for the internal carotid artery and the presence vs absence of the palatine artery (Rollot, Evers & Joyce, 2021a). While a canal for the palatine artery, the canalis caroticus lateralis, was consistently identified in most described paracryptodires (Gaffney, 1972; Brinkman & Nicholls, 1991, 1993; Brinkman, 2003), a recent review of the circulatory system of that clade revealed that the palatine artery is actually lost in most paracryptodires, with the exception of the Late Jurassic pleurosternid Uluops uluops (Rollot, Evers & Joyce, 2021a). A canalis caroticus lateralis was also recently identified in the basal baenid Lakotemys australodakotensis (Rollot et al., 2022), and a canal for the palatine artery, therefore, is unambiguously present in two paracryptodires only. Although the putative repeated independent loss of the palatine artery within paracryptodires might appear surprising at first, independent losses of that artery have been documented in other turtle clades such as testudinoids, in which it is variably present, but greatly reduced in size, or absent (Rollot, Evers & Joyce, 2021a). In addition to this new source of variability identified within paracryptodires, we here also document differences in the morphology of the carotid pit and closure of the circulatory system between members of that clade that are relevant for the understanding of the evolution of the basicranial area of paracryptodires in particular, but also have the potential to provide new insights into the evolution of the basicranium in turtles in general. In the pleurosternids Pleurosternon bullockii (Evers, Rollot & Joyce, 2020) and Uluops uluops (Rollot, Evers & Joyce, 2021a), the carotid pit is posteriorly confluent with a sulcus that extends on the ventral surface of the skull along the pterygoid-parabasisphenoid suture and which likely held the internal carotid artery. The carotid pit of Trinitichelys hiatti differs from that condition in that it is slightly smaller than that of Pleurosternon bullockii and Uluops uluops, and posteriorly constricted by extensions of the pterygoid and parabasisphenoid that partially cover the posterior aspect of the cavity and separates it from a narrow, posteriorly directed carotid groove. In Lakotemys australodakotensis, it appears that a true carotid pit and carotid groove are absent and the foramen posterius canalis nervi vidiani, foramen posterius canalis carotici lateralis, and foramen posterius canalis carotici basisphenoidalis are located in a shallow depression instead (Rollot et al., 2022). In Arundelemys dardeni, on the left side, a relatively large carotid pit is present, but is posteromedially slightly constricted by a small medial extension of the pterygoid (Evers, Rollot & Joyce, 2021), a similar morphology to that of Trinitichelys hiatti. We interpret the observed constriction along the posterior margin of the carotid pit in Arundelemys dardeni and Trinitichelys hiatti as a putative intermediate ossification stage of the basicranium between the fully open carotid pit of pleurosternids and the anteriorly completely enclosed circulatory system of more advanced baenids. This hypothesis is also supported by differences between pleurosternids and baenids regarding the ventral exposure of the foramen distalis nervi vidiani. In Uluops uluops, the foramen distalis nervi vidiani, which allows the passage of the vidian nerve from the canalis pro ramo nervi vidiani into either the carotid groove or the canalis caroticus internus, is located in the carotid groove at the posterior end of the carotid pit (Rollot, Evers & Joyce, 2021a), while in more advanced baenids, the foramen distalis nervi vidiani is not ventrally exposed and the vidian nerve, by means of the canalis pro ramo nervi vidiani, joins the canalis caroticus internus just anterior to the foramen posterius canalis carotici interni (Brinkman & Nicholls, 1991; Brinkman & Nicholls, 1993; Brinkman, 2003; Rollot, Lyson & Joyce, 2018; Rollot, Evers & Joyce, 2021a). An increased ossification of the basicranium in advanced baenids, therefore, might have led to the complete enclosure of the circulatory system and the presence of a true canalis caroticus internus and foramen posterius canalis carotici interni, ventrally covering the junction point between the canalis pro ramo nervi vidiani and internal carotid artery system.

Phylogenetic relationships

The first phylogenetic analysis under equal weighting resulted in 2,497 most parsimonious trees with 360 character-state transitions. The strict consensus tree is shown in Fig. 9A. Several polytomies are apparent at the base of the tree, at the base of baenoids, and within baenodds. The “pruned trees” function in TNT was used to identify three rogue taxa in the tree, namely Pleurosternon moncayensis, Scabremys ornata, and Goleremys mckennai. The obtained reduced most parsimonious tree has 477 character-state transitions. The new reduced strict consensus tree obtained without these taxa is shown in Fig. 9B and the time-calibrated tree with occurrence times plotted to stratigraphic time in Fig. 10. The exclusion of the aforementioned problematic taxa allowed a better resolution at the base of baenoids, with the resolution of the relationships between Riodevemys inumbragigas, Toremys cassiopeia, and Pleurosternon bullockii, and slightly improved the resolution within baenodds, by grouping Denazinemys nodosa, Boremys grandis, and Boremys pulchra together. The second phylogenetic analysis was performed under implied weighting with a K value of 12 and resulted in 10 most parsimonious trees. The strict consensus tree is shown in Fig. 11A. Pleurosternon moncayensis and Scabremys ornata were identified as problematic taxa, and the new reduced strict consensus tree obtained without them is shown in Fig. 11B. The tree with occurrence times plotted to stratigraphic time is shown is Fig. 12. The new strict consensus tree is almost fully resolved, with only one polytomy remaining between Neurankylus eximius, Neurankylus torrejonensis, and baenodds.

Figure 9 Phylogenetic hypotheses resulting from the analysis under equal weighting.

(A) Strict consensus tree obtained with all original taxa included, (B) strict consensus tree obtained without the rogue taxa Pleurosternon moncayensis, Scabremys ornata, and Goleremys mckennai.

Figure 10 Time-calibrated strict consensus tree obtained from the analysis under equal weighting, without rogue taxa.

The range of North American taxa is highlighted in blue, the range of South American taxa in yellow, the range of European taxa in red, and the range of Asian taxa in orange.

Figure 11 Phylogenetic hypotheses resulting from the analysis under implied weighting with a K factor of 12.

(A) Strict consensus tree obtained with all original taxa included, (B) strict consensus tree obtained without the rogue taxa Pleurosternon moncayensis and Scabremys ornata.

Figure 12 Time-calibrated strict consensus tree obtained from the analysis under implied weighting with a K factor of 12, without rogue taxa.

The range of North American taxa is highlighted in blue, the range of South American taxa in yellow, the range of European taxa in red, and the range of Asian taxa in orange.

The results of the two sets of analyses are overall similar with regard to the topology of Helochelydridae and Baenoidea as both of them retrieved helochelydrids as basal-branching Paracryptodira in a more inclusive position than Baenoidea. Within Baenoidea, Baenidae include Lakotemys australodakotensis, Trinitichelys hiatti, and Arundelemys dardeni as basal branching taxa and the clade is well-supported (10 synapomorphies in the reduced equally weighted strict consensus tree, 9 in the reduced implied-weighted one). Despite the subtle nature of morphological variation we identified between Arundelemys dardeni and Trinitichelys hiatti (see above), Trinitichelys hiatti is retrieved as sister taxon to Lakotemys australodakotensis at the base of Baenidae in both analyses, whereas Arundelemys dardeni is always retrieved in a less inclusive position relative to the latter two taxa within baenids. These results corroborate our suggestion to keep both taxa as separate taxonomic entities, but we note that different optimizations of shell-based characters in Arundelemys dardeni, in which the carapace and plastron are unknown, might yield to different affinities between Lakotemys australodakotensis, Trinitichelys hiatti, and Arundelemys dardeni at the base of baenids. Pleurosternidae, the sister group to Baenidae, contain Riodevemys inumbragigas, Toremys cassiopeia, and Pleurosternon bullockii. Dinochelys whitei, Glyptops ornatus, Dorsetochelys typocardium, and Uluops uluops, taxa that were often interpreted as pleurosternids, are retrieved in both analyses as basal branching paracryptodires in a paraphyletic succession outside of Baenoidea. We note that both Pleurosternidae and Baenoidea are only supported by one and two synapomorphies, respectively, and that the characters of interest are not universally preserved in the taxa retrieved as Pleurosternidae or just outside of Baenoidea (see below for further discussion about the synapomorphies). Helochelydridae are well-supported in both analyses (10 synapomorphies retrieved for the reduced equally weighted strict consensus tree vs 7 in the reduced implied weighted one), but their position as basal branching paracryptodires differs from latest insights into paracryptodiran relationships that retrieved helochelydrids deeply nested within Pleurosternidae (Rollot, Evers & Joyce, 2021a). This difference can be explained by the inclusion of additional stem testudinatans (i.e., meiolaniforms, sichuanchelyids, Eileanchelys waldmani) that likely pulled Helochelydridae outside of Pleurosternidae and Baenoidea, but still recovering them as Paracryptodira.

The main difference between the analyses presented herein is the placement of Compsemydidae, which are retrieved either as the most basal branching clade of Paracryptodira (Fig. 11B) or outside of Paracryptodira in a polytomy with Peligrochelys walshae, Chubutemys copelloi, Mongolochelys efremovi, Sichuanchelys palatodentata, and Eileanchelys waldmani. As for the placement of Helochelydridae, the addition of basal testudinatans likely dragged Compsemydidae down from the previously suggested more advanced position within baenids (Rollot, Evers & Joyce, 2021a). Despite the inclusion of basal turtles, Kallokibotion bajazidi is still retrieved as the most basal member of Compsemydidae, confirming a previous hypothesis of its phylogenetic affinities (Rollot, Evers & Joyce, 2021a), and Compsemydidae are well supported in both analyses (13 synapomorphies in the reduced equally weighted strict consensus tree, 8 in the reduced implied weighted one). In the case where Compsemydidae are not retrieved as Paracryptodira, i.e., in the reduced equally weighted strict consensus tree, Paracryptodira, with Helochelydridae as its most basal branching clade, are united by eight synapomorphies. In the reduced implied weighted strict consensus tree, only two synapomorphies unite paracryptodires with Compsemydidae as they are the most basal representatives, but the synapomorphies that unite helochelydrids with less inclusive paracryptodires are nearly identical to those in the former analysis. Future studies that include a larger sample of basal turtles and possibly basal Testudines are expected to clarify the phylogenetic placement of Compsemydidae and its affinities with Paracryptodira.

As already mentioned above, our analyses consistently retrieved the same topologies for Helochelydridae and Baenoidea, and the synapomorphies uniting Helochelydridae, Baenoidea, Baenidae, and Pleurosternidae are nearly if not fully identical. Baenidae are well-supported by at least nine synapomorphies in both the equally weighted and implied weighted strict consensus trees: a reduced lingual ridge, only developed anteriorly (character 5, state 1; absent in Palatobaena spp.), a ventral extension of the jugal (character 18, state 0; absent in some baenids but mostly due to intraspecific variation), a well-developed posterior process of the pterygoid with an extensive contact with the basioccipital (character 26, state 3), nasals that extend as far anteriorly as the premaxillae (character 68, state 0; nasal do not extend as far anteriorly as the premaxillae in baenodds), present and extensive contact of the axillary buttresses with the costals (character 84, state 1), present and extensive inguinal buttresses (character 85, state 1), a deep upper temporal emargination relative to the incisura columella auris (character 95, state 2), absence of basipterygoid processes (character 96, state 2), and absence of a second pair of anterior tubercula basioccipitale (character 97, state 0). A single additional synapomorphy is found in the equally weighted strict consensus tree and not the implied weighted one, the absence of epiplastral processes (character 45, state 1). The presence of extensive axillary and inguinal buttresses, the elongate pterygoid-basioccipital contact, and the absence of epiplastral processes are features that have been commonly used to distinguish Baenidae from Pleurosternidae within Baenoidea, and retrieving them as synapomorphies for Baenidae therefore does not come as a surprise. The character pertaining to the depth of the upper temporal emargination was modified recently (see Rollot, Evers & Joyce, 2021a) in order to take into account more precisely the extent of the upper emargination, and our analyses retrieved a deep upper temporal emargination as a synapomorphy of baenids (with the exception of Hayemys latifrons and Palatobaena gaffneyi). All the other turtles in our dataset are characterized by an upper emargination that is shallow to absent, with the exception of Pleurosternon bullockii, in which it is moderate and, in lateral view, ends between the anterior margin of the cavum tympani and the incisura columella auris. Similarly, the absence of a basipterygoid process unambiguously characterizes all baenids, which contrasts with the condition observed in pleurosternids and basal branching paracryptodires just outside of Baenoidea, in which the basipterygoid process is still present albeit reduced. The presence vs absence of anterior tubercula basioccipitale has recently been introduced as a character in matrices built to investigate paracryptodiran relationships (Rollot, Evers & Joyce, 2021a), and the tubercula are unambiguously absent in baenids, but present in all other paracryptodires with the exception of Compsemydidae. The extension of the nasals as far anteriorly as the premaxillae is retrieved as a synapomorphy for Baenidae, but this feature is only observed in basal baenids, as baenodds but also various basal branching paracryptodires and pleurosternids have nasals that do not extend as far anteriorly as the premaxillae. Finally, baenids are also united by a reduced lingual ridge (complete in Peckemys brinkman; absent in Palatobaena spp.), but this character seems to be somewhat more variable outside of Baenidae in our sample.

The clades Pleurosternidae and Baenoidea are both weakly supported in our analyses. Pleurosternidae are only supported by one synapomorphy, wider than long vertebrals (character 39, state 0), while Baenoidea are united by two synapomorphies, a moderately deep upper temporal emargination relative to the incisura columella auris (character 95, state 1) and the absence of a canal for the palatine artery (character 100, state 1). Our analyses retrieved exclusively cranial characters as synapomorphies for Baenoidea, but the pleurosternids Riodevemys inumbragigas and Toremys cassiopeia are actually solely known by shell material. These characters could, therefore, not be originally scored in our matrix for the aforementioned taxa, and were optimized a posteriori by the software as synapomorphies for Baenoidea despite the lack of information for Riodevemys inumbragigas and Toremys cassiopeia. A similar issue affects the placement of Uluops uluops, a taxon that was commonly found as a member of Pleurosternidae in phylogenetic analyses of paracryptodiran relationships, but that is retrieved as a basal branching paracryptodire just outside of Baenoidea in our analyses. The dimensions of the vertebral scutes, of which character state 0 (wider than long) is identified as the unique synapomorphy of Pleurosternidae, remain unknown for Uluops uluops as this taxon is exclusively known by cranial material. It is therefore likely that small changes in the matrix could bring some of the taxa retrieved just outside of Baenoidea into Pleurosternidae, and vice versa, and the addition of stem Testudinata in our matrix might have dragged down some paracryptodiran taxa, again, which results in Pleurosternidae to only include three taxa in our analyses.

Our analyses recovered Helochelydridae as basal branching paracryptodires in a more inclusive position than Dinochelys whitei, Glyptops ornatus, and the clade formed by Dorsetochelys typocardium and Uluops uluops. We retrieved 10 synapomorphies that unite helochelydrids in the equally weighted strict consensus tree vs seven in the implied weighted strict consensus tree. The seven synapomorphies retrieved in the implied weighted strict consensus tree are all also found in the equally weighted strict consensus tree, and those common synapomorphies are as follows: a large contribution of the frontal to the orbit (character 15, state 2), absence of a labial ridge on the mandible (character 70, state 1), presence of articular surfaces on the cervical vertebrae (procoelous or opisthocoelous; character 76, state 1), a shell sculpturing made of raised tubercles that easily break (character 78, state 2), presence of three suprapygals (character 86, state 2), presence of an entoplastral scute (character 102, state 1), and presence of V-shaped anterior peripherals (character 103, state 1). The three additional synapomorphies in the equally weighted strict consensus tree are the absence of epiplastral processes (character 45, state 1), large mandibular condyles (character 60, state 1), and a large external narial opening (character 73, state 1). The frontal contribution to the orbit appears to be quite variable in our dataset. The absence of a labial margin on the mandible seems, at first glance, to be a relatively solid character among our sample as all turtles in a more basal position than helochelydrids and for which this character can be scored, actually possess a labial margin on the mandible. However, we note that this feature is homoplastically shared with some baenids and that lots of missing data (about 50%) remain in our dataset for this character. The presence of articular surfaces on the cervical vertebrae is even more poorly characterized in our matrix, as this feature remains unknown for 75% of the taxa used in our study. On the other hand, features such as the sculpturing of the shell made of raised tubercles, presence of three suprapygals, presence of an entoplastral scute, and presence of V-shaped anterior peripherals are robust and reliable, as they are diagnostic of that clade (Joyce, 2017).

Compsemydidae, including Kallokibotion bajazidi as its most basal representative, are alternatively found outside of Paracryptodira in a polytomy with basal testudinatans (reduced equally weighted strict consensus tree, Fig. 9B) or as the most basal branching clade of Paracryptodira (reduced implied weighted strict consensus tree, Fig. 11B). In the reduced equally weighted strict consensus tree, Compsemydidae are supported by 13 synapomorphies, of which eight also support this clade in the implied weighted strict consensus tree. The synapomorphies common to both analyses are as follows: absence of a posterodorsal extension of the quadratojugal (character 19, state 1), presence of an elongate process of the pterygoid with an extensive contact with the basioccipital (character 26, state 3), absence of cervical scutes (character 38, state 0), absence of a nuchal contribution to the anterior margin of the shell (character 80, state 1), presence and extensive contact of the axillary buttresses with the costals (character 84, state 1), presence of extensive inguinal buttresses (character 85, state 1), absence of a contact between peripheral I and costal I (character 90, state 0), and the superficial enclosure of the cavum tympani by a squamosal-quadrate contact (character 105, state 1). The following synapomorphies additionally unite Compsemydidae in the equally weighted strict consensus tree only: a long preorbital length (character 2, state 1), a midline contact of the pterygoid of 40–70% of their length (character 25, state 2), the anterior margin of marginal I mainly located over peripheral I (character 44, state 2), the absence of epiplastral processes (character 45, state 1), and an entoplastron with equal dimensions (character 93, state 1; becomes broader than long in more derived compsemydids). The absence of a posterodorsal extension of the quadratojugal is not unique to compsemydids as it is homoplastically shared with Sichuanchelys palatodentata, Baena arenosa, and Chisternon undatum. However, little is actually known with certainty about the cranial anatomy of the latter two taxa, and this feature therefore appears to be a robust synapomorphy in the context of our analysis. The presence of an elongate process of the pterygoid is shared with baenids, but clearly distinguishes compsemydids from other basal turtles as these latter taxa lack such a posterior extension of the pterygoid and extensive contact with the basioccipital. The condition observed in Baenidae and Compsemydidae, therefore, appears to be more advanced than any other stem turtle in our dataset. Similarly, compsemydids share with baenids the presence of extensive axillary and inguinal buttresses. The aforementioned three features were found as synapomorphies of Baenidae including Compsemydidae in recent analyses (see Rollot, Evers & Joyce, 2021a), but as already mentioned above, the inclusion of additional basal testudinatans likely pulled Compsemydidae down outside of Baenidae, leading to a different result herein and the homoplastic sharing of those features between the two clades. The absence of cervical scutes distinguishes compsemydids from nearly any other turtle in our dataset, and it appears to be a relatively solid character to unite Compsemydidae, although we note that two pleurosternids, Pleurosternon bullockii and Toremys cassiopeia, also lack cervical scutes. The absence of a nuchal contribution to the anterior margin of the shell, absence of a contact between peripheral I and costal I, and superficial enclosure of the cavum tympani by a squamosal-quadrate contact are unique to Compsemydidae and represent robust synapomorphies for the clade. The placement of Kallokibotion bajazidi within Compsemydidae was already strongly supported in a previous analysis of paracryptodiran relationships (Rollot, Evers & Joyce, 2021a), but this result needed to be tested in a global context as the study of interest did not include non-paracryptodiran taxa, with the exception of the outgroups Proganochelys quenstedti and Kayentachelys aprix. Despite the inclusion of Meiolaniformes, Sichuanchelyidae, and Eileanchelys waldmani in a modified version of the matrix used by Rollot, Evers & Joyce (2021a), Kallokibotion bajazidi is still retrieved as a basal Compsemydidae herein. This relationship remains strongly supported and holds true independently of the placement of compsemydids within Paracryptodira or not, showing that the inclusion of Kallokibotion bajazidi within Compsemydidae is robust.

The varying placement of Compsemydidae in our analyses affects the synapomorphies retrieved as uniting Paracryptodira, but most of the synapomorphies of the latter found in the analysis that recovers Helochelydridae as the most basal branching paracryptodiran clade still unite helochelydrids and less inclusive paracryptodires in the analysis under implied weighting, where Compsemydidae are retrieved as Paracryptodira. In the equally weighted strict consensus tree, Paracryptodira is found within a polytomy with several basal turtles and Compsemydidae, and united by eight synapomorphies: an elongate skull shape (character 1, state 1), absence of a lingual ridge (character 5, state 2; present but reduced in most baenids), a small contribution of the frontal to the orbit (character 15, state 1), absence of a jugal contribution to the orbit (character 17, state 2; but small to absent in several taxa), absence of a ventral process of the jugal (character 18, state 1; present in most baenids), a Z-shaped dentary-surangular contact (character 31, state 0), presence of a distinct skull sculpturing (character 77, state 1; absent in derived baenids), and presence of a pair of anterior tubercula basioccipitale on the parabasisphenoid (character 97, state 1; absent in baenids). The extent of the contribution of the frontal to the orbit appears to be highly variable within paracryptodires. A small contribution is retrieved as a paracryptodiran synapomorphy, but this contribution is actually large in helochelydrids, the most basal branching paracryptodiran clade in our analysis. Other less inclusive basal branching paracryptodires and pleurosternids indeed have a small contribution of the frontal to the orbit, but all character states are variably present in baenids, and this feature therefore appears not to be a good diagnostic character for paracryptodires. The absence of a jugal contribution to the orbit is retrieved as synapomorphic of paracryptodires, and contrasts from the condition observed in all other basal testudinatans including Compsemydidae, in which the jugal contribution to the orbit is large. The jugal of helochelydrids is indeed excluded from the orbital margin, but this condition is not universally found in other basal branching paracryptodires. Within Baenoidea, the jugal is again excluded from the orbital margin in Pleurosternidae and the most basal baenids, but small to large contribution or a complete absence of a contribution to the orbital margin are all found within advanced baenids. This feature thus also appears not to be a reliable diagnostic character. Similarly, the ventral process of the jugal is absent in pleurosternids and basal branching paracryptodires (polymorphic in Naomichelys speciosa), and present in most baenids (also polymorphic in some baenodds), but this character could only be scored for one third of the non-baenid taxa retrieved as paracryptodires herein, and the interpretation of this character therefore needs to be taken with caution. Likewise, the shape of the dentary-surangular contact remains unknown in more than two thirds of all taxa retrieved as paracryptodires herein, so that caution is required for this character as well. On the other hand, the four other synapomorphies identified in the strict consensus tree seem to be more robust and less variable within the tree. The skull of all basal branching paracryptodires (with the exception of Uluops uluops), pleurosternids, and the two most basal baenids Lakotemys australodakotensis and Trinitichelys hiatti is elongate, and change in the skull shape appears to occur with Arundelemys dardeni, of which the skull is more wedge-shaped as all more advanced baenids (although considered as rounded in our matrix for Palatobaena spp., but still not elongate). The lingual ridge is completely absent in all basal branching paracryptodires (again, with the exception of Uluops uluops) and pleurosternids, but is present albeit reduced in most baenids (absent in Palatobaena spp.). Anterior tubercula basioccipitale are present in all basal branching paracryptodires and pleurosternids and appear to be subsequently lost in baenids. Finally, all basal branching paracryptodires, pleurosternids and basal baenids have their bone surface decorated by a distinct, albeit variable, sculpturing which tends to disappear towards more advanced baenids.

In the implied weighted strict consensus tree, Paracryptodira includes Compsemydidae as its most basal branching clade but is only united by two synapomorphies, i.e., a moderately long posterior process of the pterygoid with a point contact with the basioccipital (character 26; state 2; this contact with the basioccipital becomes independently elongate in compsemydids and baenids) and an entoplastron about as wide as long (character 93, state 1). The extent of the posterior process of the pterygoid is actually variable within Paracryptodira. In Compsemydidae, i.e., the most basal branching clade of paracryptodires in this analysis, the posterior process of the pterygoid is well developed and has an elongate contact with the basioccipital. In Helochelydridae, the following less inclusive clade within Paracryptodira, the posterior morphology of the pterygoid is greatly variable, as a point contact with the basioccipital occurs in Helochelydra nopcsai, whereas this contact is completely absent in Naomichelys speciosa, and the condition remains unknown in Aragochersis lignitesta. The posterior process of the pterygoid is moderately developed in other less inclusive basal branching paracryptodires, as well as in Pleurosternidae (although unclear in Pleurosternon bullockii), while Baenidae homoplastically exhibit the same condition as Compsemydidae. We, however, note that the change in the posterior extent of the pterygoid along the turtle stem still highlights an intriguing evolutionary trend. In most of the basal testudinatans included in our study, the pterygoid lacks an extended posterior process and the prootic is apparent in ventral view. In Mongolochelys efremovi, for instance, the posterior process of the pterygoid is more developed than in other basal Testudinata and the prootic is concealed in ventral view, but a posterior contact with the basioccipital is prevented by a small posterolateral lappet of the parabasisphenoid. A more advanced condition is then observed in pleurosternids and some basal paracryptodires, where a pterygoid-basioccipital contact is apparent, and this contact becomes even more extensive in baenids and compsemydids, for most of or even the entire length of the basioccipital plate. The second synapomorphy, i.e., an entoplastron that is as wide as long, is even more variable within Paracryptodira than the other one. The entoplastron is indeed as wide as long in Kallokibotion bajazidi but broader than long in other compsemydids. Similarly, in helochelydrids, the entoplastron is as wide as long in Aragochersis lignitesta but longer than broad in Helochelydra nopcsai and Naomichelys speciosa. In the less inclusive basal branching paracryptodires Dinochelys whitei and Glyptops ornatus, the entoplastron is as wide as long. Within Baenoidea, the entoplastron is broader than long in pleurosternids (although polymorphic and sometimes as wide as long in Pleurosternon bullockii), and in baenids, the entoplastron is variably either longer than broad or as wide as long, and this character could not be scored for about half of them. At the end, the synapomorphies retrieved in the analysis under equal weighting, in which Paracryptodira do not include Compsemydidae, appear to be more robust and reliable considering our knowledge about the evolutionary history of paracryptodiran sub-clades (i.e., skull shape, anterior tubercula basioccipitale, skull sculpturing).

The matrix used herein includes the most recent and novel insights into the cranial and shell anatomy of paracryptodires. As we included several basal testudinatans in our sample, the placement of Kallokibotion bajazidi within Compsemydidae and that of Helochelydridae as basal branching Paracryptodira, are robust to the inclusion of taxa that represent previously proposed close relatives outside of Paracryptodira. Other aspects, such as the synapomorphies of Paracryptodira, Baenoidea, and Pleurosternidae, or the taxonomic composition of Pleurosternidae, are dependent on the taxonomic content of these groups, as the synapomorphies are changing when different taxa are added to a pre-defined clade.

A similar issue was highlighted by the recently published study of Tong et al. (2022), in which clear synapomorphic characters were not retrieved for Paracryptodira and Baenoidea despite the use of a different matrix. One of the analyses provided by the aforementioned study also yielded an alternative phylogenetic hypothesis with the placement of Compsemydidae as Pan-Pleurodira and that of Kallokibotion bajazidi within Pleurosternidae. Such affinities were not retrieved in the present study, but we acknowledge that the inclusion of representatives of the turtle crown and putative stem groups (e.g. Thalassochelydia, Sinemydidae, Macrobaenidae, and Xinjiangchelyidae) to our matrix might impose more constraints on some parts of the tree and provide alternative phylogenetic hypotheses. A redescription of the skulls of Dinochelys whitei, Glyptops ornatus and Dorsetochelys typocardium, which are also included in the phylogenetic analyses of Tong et al. (2022), is also expected to provide new insights into the cranial anatomy of these taxa and, consequently, might allow additional characters to be scored and more precisely identify the synapomorphies of Baenoidea and Pleurosternidae. Finally, the study of phylogenetic affinities of paracryptodires in a global context should be accompanied by a specific focus on Eileanchelys waldmani and Heckerochelys romani, two of the oldest known aquatic turtles that lived during the Middle Jurassic (Sukhanov, 2006; Gordenko, 2008; Anquetin et al., 2009; Anquetin, 2010). Pleurosternidae are known to be aquatic and were relatively widespread along the Jurassic-Cretaceous boundary both in North America and Europe, but study of fragments has shown that representatives of the group were likely present in the Middle Jurassic of England (Scheyer & Anquetin, 2008). Given that Eileanchelys waldmani was found in Scotland and several pleurosternid taxa were collected around the Jurassic-Cretaceous boundary in western Europe including England, putative close relationships between all aquatic taxa of that area have to be considered in future studies. Heckerochelys romani is known by dozens of fragments, but only a few have been described and illustrated, and a reanalysis of all the available material, including µCT scans of the skull material, could provide new insights into the evolution and diversification of aquatic turtles.

Conclusion

Here we provide the first detailed description of the skull of the Early Cretaceous (Aptian-Albian) baenid Trinitichelys hiatti. We propose an alternative interpretation of the cranial osteology of Trinitichelys hiatti to the original publication and highlight many similarities with the recently described basal baenid skulls of Arundelemys dardeni and Lakotemys australodakotensis. Arundelemys dardeni and Trinitichelys hiatti appear to be nearly identical and can only be distinguished by minor differences. Additional insights, particularly regarding the anatomy of the posterior portion of the skull and the age of both taxa, will allow future clarification if synonymizing the latter two names is warranted. Trinitichelys hiatti also shares with Arundelemys dardeni and Lakotemys australodakotensis an intriguing combination of features that are typical of both Pleurosternidae or Baenidae. The circulatory system of Trinitichelys hiatti, especially the morphology of the carotid pit, is suggested to represent an intermediate condition between that of pleurosternids (e.g., Uluops uluops) and baenids (e.g., Eubaena cephalica). We also provide an expanded phylogenetic analysis that includes a selection of stem turtles and latest described and valid paracryptodiran species. We find Trinitichelys hiatti as sister taxon to Lakotemys australodakotensis at the base of Baenidae and retrieve Helochelydridae along the stem of Baenoidea, but recover Dinochelys whitei, Glyptops ornatus, Dorsetochelys typocardium, and Uluops uluops as basal branching Paracryptodira. The study of paracryptodiran relationships in a global context along with some of the oldest known aquatic stem turtles is expected to provide new insights into the evolution of paracryptodires and the diversification of an aquatic lifestyle in Testudinata.

Supplemental Information

Supplemental Information 1 Scorings changed from the matrix of Rollot et al. (2021).

Click here for additional data file.

Supplemental Information 2 Matrix.

Click here for additional data file.

We are grateful to Christina Byrd and Megan Whitney at the Museum of Comparative Zoology for assistance with specimen access and µCT scanning. We also thank the editor Fabien Knoll and the reviewers Donald Brinkman, Heather Smith, and Julien Claude for helpful comments and suggestions that improved the quality of the manuscript.

INSTITUTIONAL ABBREVIATIONS

DORCM Dorset Museum, Dorchester, England

MCZ Harvard Museum of Comparative Zoology, Cambridge, Massachusetts, U.S.A

DMNH Denver Museum of Nature and Science, Denver, Colorado, U.S.A

Additional Information and Declarations

Competing Interests

Author Contributions

Data Availability

Stephanie E. Pierce is a PeerJ Academic Editor. The authors declare that they have no competing interests.

Yann Rollot conceived and designed the experiments, performed the experiments, analyzed the data, prepared figures and/or tables, authored or reviewed drafts of the article, and approved the final draft.

Serjoscha W. Evers conceived and designed the experiments, performed the experiments, analyzed the data, authored or reviewed drafts of the article, and approved the final draft.

Stephanie E. Pierce analyzed the data, authored or reviewed drafts of the article, and approved the final draft.

Walter G. Joyce conceived and designed the experiments, performed the experiments, analyzed the data, authored or reviewed drafts of the article, and approved the final draft.

The following information was supplied regarding data availability:

The matrix is available in the Supplemental File.

The 3D models and CT scans used for this study are available at MorphoSource: Project ID is 000417478.

https://www.morphosource.org/projects/000417478?locale=en.

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
