# Peer review of "Cranial osteology, taxonomic reassessment, and phylogenetic relationships of the Early Cretaceous (Aptian-Albian) turtle Trinitichelys hiatti (Paracryptodira)"

_PeerJ, doi:10.7717/peerj.14138_

## Round 0.1 · original submission · Minor Revisions

Please, consider including as a separate document the complete list of characters with coding. Additional figures (e.g., caudal view of the skull in fig. 6) would be much appreciated too.

·

Basic reporting

the paper is clear and well written. I noted two or three typos or editorial issues:
line 109: space missing between "additional changes"

line 344-345: "reminds of the septum" is missing a word, eg: "reminds us of the septum" or "is reminiscent of the septum"

line 1384: "allow to score" is missing a word, eg: "allows us to score" or "allows additional characters to be scored"

Experimental design

no comment

Validity of the findings

no comment

Additional comments

A great paper.

·

Basic reporting

This is a significant paper worthy of publication. Trinitichelys hiatti falls at an important temporal and phylogenetic position, and the study skull is the only one currently known for the species. Thus, its morphology is of particular importance not only for understanding T. hiatti, but for reconstructing the evolutionary history of the speciose baenid clade. The authors use microCT data to segment osteological features and individual cranial bones, which enables a more accurate description of the morphology of the skull than was previously possible during its initial study in the 1970s. This new technology reveals phylogenetically and functionally relevant differences compared to the original descriptions of the skull, which have important implications for paracryptodire evolution.

I commend the authors on conducting such a strong study. The descriptions are extremely well-written and detailed, and the figures are clear. There are a few regions in which the morphology is not able to be definitively determined (e.g., epipterygoid, contacts among some bones), and I appreciate the authors’ conservative interpretation and their circumspect discussion of these regions. In all, this is an excellent paper.

I have only one major request (in Section 3 below). Everything else should be considered a minor suggestion or query.

Minor typographical and grammatical suggestions/edits:
Is it a PeerJ formatting standard to italicize family names? If not, I’d suggest reverting them to normal text.
Line 88: “pleuorsternid” should be “pleurosternid”
Line 109: Missing space between “additional” and “changes”
Line 116: At the beginning of a sentence, suggest writing out a number in words (i.e., “One thousand” rather than “1000”).
Line 149: Suggest changing “anteriorly narrower than posteriorly” to “narrower anteriorly than posteriorly”.
Line 184: The word “margin” is used twice in rapid succession. Could one of these be changed (e.g., “posterior aspect of the orbit margin”)?
Line 217: Suggest changing “Does not allow to assess precisely its general shape” to “does not allow its general shape to be precisely assessed”.
Lines 221-227: Is it necessary to mention three times that the nasal contacts its counterpart medially? I recognize that you’re listing different views, but perhaps for the dorsal view it would be simpler to say, “In dorsal view, the nasal also contacts the anterior process of the frontal…” and omit the repeated phrase about midline nasal contact.
Figure 1. The abbreviation “ju” (presumably jugal) isn’t listed on the figure legend.
Lines 734, 758, and Figure 2 Legend (elsewhere?): For consistency with the formal anatomical terminology used elsewhere throughout the manuscript, I wonder if the authors want to use “condylus occipitalis” rather than “occipital condyle”?
Line 879: Suggest changing “than what Gaffney inferred” to “than Gaffney inferred”
Line 964: Suggest changing “great nasal variation to not be of taxonomic importance” to
“great nasal variation to be of no (or limited or minimal) taxonomic importance”
Line 1021: Suggest changing “allow to determine” to “allow a determination of”
Lines 1322-3: “taken caution” should be “taken with caution”.
Line 1363: “as wide as long as wide” should be “as wide as long”.
Line 1384: Suggest changing “allow to score” to “allow the scoring of”
Line 1388: Suggest changing “to help clarifying” to “to help clarify”
Line 1413: Suggest changing “will allow clarifying in the future” to “will allow future clarification”

Experimental design

Other queries and suggestions:
Lakotemys: The comparative anatomy of this taxon is particularly important for the current paper given the sister relationship between the two. I recognize that it’s tricky when authors have two related papers under consideration or in press at the same time. However, as I’m writing this review, the Rollot et al. (in press) paper about Lakotemys is not currently published, which it makes it difficult to assess the comparisons to this apparent sister taxon. I don’t fault the authors for this, and I don’t know that there is a current solution, so I’ll just note that I am taking the authors’ word for the apparent similarities between the two taxa any time that the Rollot et al. (in press) paper is cited. I assume this paper will be available soon.

Mandible: Gaffney always referred to the mandible in turtles as the “lower jaw”, presumably due to its composition of multiple bones and difference in composition from the archetypal mammalian mandible. I was also a bit surprised to see the term used here as a plural, “mandibles” (line 129). Could the authors please briefly explain how they use the term “mandible” and their rationale for using it over “lower jaw”? Same question for stapes vs. columella.

Line 113: Which version of TNT was used?

Lines 163-5: It is indicated that 4070 skull is longer than wide. However, the sentence “Therefore, it is less elongate than Glyptops” doesn’t quite follow. Do the authors mean the proportions of length to width are comparable between Pleurosternon and 4070, therefore indicating that 4070 is less elongate than Glyptops? Please rephrase for clarity.

Lines 178-189: The discussion of cranial scute sulci is helpful and interesting. More helpful still would be if they were labeled on Figure 1.

Comparative anatomy: As the authors well know, the fossil record of baenids is rich and well-documented. In the comparative anatomy section, were all taxa with available morphology mentioned for comparison in each region? If not, how were the representatives chosen? For example, I didn’t see much mention in this section of Arvinachelys, Stygiochelys, Baena arenosa, Chisternon, Boremys, or Neurankylus.

Lines 225-227: The anterior process of the frontal “slightly protruding” between the nasals is not apparent in Figure 1. At least, no such protrusion is apparent on the left in Figure 1. Perhaps the CT scans show more protrusion? I agree that the nasals appear to be in contact with each other for essentially the entirety of their length.

Lines 402-3: A wise scientist (possibly Walter Joyce?) once advised me against inferring muscular attachments in fossil turtle specimens. This one also stands out as no other muscular attachments are mentioned in the descriptions. Consider omitting or adding information about other muscle attachments.

Line 416: Can you please elaborate briefly on the “tongue groove”? Is this the inferred location where the tongue rested against the palate anteriorly during the animal’s life?

Line 436-8: As the authors note, the dimensions of the ascending process of the maxilla are “highly unusual” for a Paracryptodire. Is the ascending process exposed bilaterally on the skull roof? I assume so, but it’s not entirely clear from Figure 1. If it’s bilateral, then certainly you can have more confidence in the accuracy of this unusual finding. It seems that the shapes of the prefrontal and the lateral aspect of the frontal around the orbit are also a bit unusual. In fact, the arrangement of the bones surrounding the dorsal margin of the orbit is quite unique. I can understand why the original Gaffney reconstruction was greatly simplified in comparison. The presumed sulcus contacting the orbit anterodorsally could easily be mistaken for the frontal suture in the absence of CT data. I appreciate that the authors provided the supporting CT data so that these unexpected findings can also be independently verified.

Line 538-546: Was the carotid pit first described by Rollot et al. (2021)? Has it been described in any taxa other than Uluops?

Lines 640, 921: The authors mention a more prominent processus trochlearis oticum in T. hiatti than Arundelemys. However, I’m having a hard time seeing the pto clearly on any of the figures. I recognize that this region can be a bit difficult to demonstrate in standard anatomical views depending on the degree of temporal emargination, but it would be nice to see this difference. Do the authors have any concerns with using this trait to differentiate among taxa given that it can be affected by sexual dimorphism?

Line 677: Do you mean that the opisthotic contacts the quadrate anterolaterally?

Lines 718-720: This sentence is a bit unclear. What is the “it” at the end of the sentence? The skull roof?

Lines 768-774: The written description of the relationship of features on the parabasisphenoid is excellent. However, it’s a bit difficult to make out some of these features in Figure 5A, which is cited parenthetically. Could corresponding labels (e.g., retractor bulbi pits, dorsum sellae, sella turcica) be added to the figure for clarity?

Lines 774-5: If the clinoid processes are missing, what is the basis for the interpretation that they “likely had broad bases…”. I’m sure the authors have a good reason for this interpretation, so it would be helpful to state it here.

Line 916: Rather than “differently shaped”, it would be helpful to remind the reader here exactly how the nasals are differently shaped between the taxa.

Lines 931-1005: This section of the Discussion relies heavily on two papers by Bever et al. 2009. Do Bever’s studies focus exclusively on Pseudemys texana and Sternotherus odoratus? Given that these taxa are quite distantly related to baenids, I wonder about the extent to which resulting inferences can be applied here. Are there any other similar studies that could be referenced to confirm that the patterns observed in the Bever studies aren’t specific to Pseudemys or emydids?

Lines 1047-1049: I think a few more references would be helpful here to support the fact that the circulatory system “has been extensively used for decades”.

Phylogenetic results: I wonder if the authors could speak to their decision to use strict consensus trees, which can be quite conservative. Were semi-strict or majority rule trees generated? If so, was there resolution in any of the unresolved polytomies?

Validity of the findings

Excellent. Superb study!

Major request:
Could the authors please consider publishing the entire character list including descriptions of character states and coding, not just the data matrix? I understand that various iterations of this character list have been previously published, but it can be challenging for the reader to find the character list spread across several papers. For example, to find the full character list here, I had to refer to Rollot et al. 2021, then Joyce & Rollot (2020), Lyson et al. (2019), and finally Lyson et al. (2016) where I found a full version of the character list, albeit minus the subsequent modifications from the above studies. Long story short, having the entire character list associated directly with each paper (supplementary is fine) would be immensely appreciated. Thank you in advance for considering this request.

Additional comments

The authors are welcome to know my identity and to contact me directly with any questions.

Sincerely,
Heather Smith

·

Basic reporting

The description is well done, and this work adds a lot of information in addition to what was published on that genus. It provides important comparison with related species. This is an important contribution for understanding the evolution and anatomy of mesozoic turtles. There is also an interesting discussion regarding the variability of cranial characters and potential taxonomic issues with other similar turtles that I was appreciating. I think that the paper could greatly benefit some additional figures and more annotations of anatomical features, and that it could be updated with a in press paper recently reviewed by some of the authors here.

My remarks are relatively minor to very minor. The paper is already very good.

1. Figures
1.1.additional figures welcome:
It would be good for the reader (although I understand you might not have it easily) to extract from your scan a close up and an interpretation of the pterygoid/quadrate/exoccipital area in posterior view with labels (cavum acustico jugulare, eventual foramina etc: see figure 85 to 101 in gaffney compendium 1979 for examples ). This part is not annotated in any of your figures so far, but you comment it in the text. The posterior view of the skull is not detailed in Gaffney 1972, and I think you could make based on the scan you obtained.
1.2. the position of the tuberculum basioccipitale should be indicated on the figures.
1.3. the hyoid apparatus discussed in the text should be illustrated.

2. phylogenetic analysis

2.1. equal weighting. What does that mean regarding ordered characters: where they scaled so that they would never count for more than 1 character change ?

2.2. You should maybe consider that the topology rather to be confirmed, might be easily challenged by including more "modern" like taxa, that can indeed impose important constraints to some part of the tree (and you do not know if these more advanced taxa are "more advanced" or not).

2.3. As some of you have reviewed the attached paper and as it is now in press, it would be wise to discuss your results by comparison to that study too (see attached ms). I do not necessarily ask to do all the analyses again as your conclusions regarding synapomorphies are already very interesting, but the alternative hypotheses obtained in the attached ms might be interesting to compare (even shortly). As I said before (and as you also say in your discussion, your analysis is not including some other well diversified clades (pleurodira, cryptodira, plesiochelyids, etc) and the results you obtain might come from the fact that this part of the tree is unconstrained with these taxa and has some taxonomic bias. You say that incorporating these taxa could help to clarify, but maybe it could make the interpretation of many synapomorphies more complicated as iterative evolution might obscure homology patterns.

2.4. Part of your conclusion regarding the variability of characters and validity of Arundelemys seems to be verified by your phylogenetic hypotheses (this is not discussed).

2.5. The parsimony scores of weighted analyses are useless: they cannot be compared with anything; however, the score of the reduced tree (once you have prunned some taxa in your analysis 1), should be provided.

3. diagnosis
in the diagnosis, to justify it is a baenid: try to put skull characters together, and postcranial characters together to make it clearer.

4. Loss of palatine artery (matter of a discussion in the text or not)... That subject might be discussed a bit more as you are reasoning here on fossils and the absence of lateral carotid canal does not necessarily involves the loss of the artery.

1249: present -> presence

1357. testudinatans -> not sure whether we should employ this neologism (it is indeed appearing also elsewhere but is it exact), Testudinata might be more exact.

Overall, this publication is very interesting, I hope that my comments will be useful.

This is signed: Julien CLAUDE (Institut des Sciences de l'Evolution de Montpellier)

Experimental design

NA

Validity of the findings

NA (see basic reporting)

Additional comments

see basic reporting

---

## Round 0.2 · accepted · Accept

Please only italicize the names of species and genera.

·

Basic reporting

The writing is excellent and meticulous.

Experimental design

No suggested improvements.

Validity of the findings

The study will be an impactful contribution to the field.

Additional comments

Brilliantly done! The authors have fully addressed my previous concerns, and I have no further suggestions. I congratulate the team on an excellent study and paper. I look forward to seeing it published.